# Host Vesicle Fusion Protein VAPB Contributes to the Nuclear Egress Stage of Herpes Simplex Virus Type-1 (HSV-1) Replication

**DOI:** 10.3390/cells8020120

**Published:** 2019-02-03

**Authors:** Natalia Saiz-Ros, Rafal Czapiewski, Ilaria Epifano, Andrew Stevenson, Selene K. Swanson, Charles R. Dixon, Dario B. Zamora, Marion McElwee, Swetha Vijayakrishnan, Christine A. Richardson, Li Dong, David A. Kelly, Lior Pytowski, Martin W. Goldberg, Laurence Florens, Sheila V. Graham, Eric C. Schirmer

**Affiliations:** 1The Wellcome Centre for Cell Biology, University of Edinburgh, Edinburgh EH9 3BF, UK; nsaizros@gmail.com (N.S.-R.); Rafal.Czapiewski@ed.ac.uk (R.C.); s1581423@sms.ed.ac.uk (C.R.D.); Dario.Barreiros@ed.ac.uk (D.B.Z.); David.Kelly@ed.ac.uk (D.A.K.); lior.pytowski@gmail.com (L.P.); 2MRC-University of Glasgow Centre for Virus Research, Institute of Infection, Immunity and Inflammation, College of Medical, Veterinary and Life Sciences, University of Glasgow, Glasgow G61 1QH, UK; I.Epifano.1@research.gla.ac.uk (I.E.); Andrew.Stevenson.2@glasgow.ac.uk (A.S.); Marion.McElwee@gla.ac.uk (M.M.); Swetha.Vijayakrishnan@glasgow.ac.uk (S.V.); l.dong.1@research.gla.ac.uk (L.D.); 3Stowers Institute for Medical Research, Kansas City, MO 64110, USA; ses@stowers.org (S.K.S.); LAF@stowers.org (L.F.); 4School of Biological and Biomedical Sciences, Durham University, Durham DH1 3LE, UK; a.c.richardson@durham.ac.uk (C.A.R.); m.w.goldberg@durham.ac.uk (M.W.G.)

**Keywords:** nuclear envelope, nuclear membrane protein, herpes simplex virus-1 (HSV-1), vesicle fusion protein (VFP), VAPB, nuclear egress

## Abstract

The primary envelopment/de-envelopment of Herpes viruses during nuclear exit is poorly understood. In Herpes simplex virus type-1 (HSV-1), proteins pUL31 and pUL34 are critical, while pUS3 and some others contribute; however, efficient membrane fusion may require additional host proteins. We postulated that vesicle fusion proteins present in the nuclear envelope might facilitate primary envelopment and/or de-envelopment fusion with the outer nuclear membrane. Indeed, a subpopulation of vesicle-associated membrane protein-associated protein B (VAPB), a known vesicle trafficking protein, was present in the nuclear membrane co-locating with pUL34. VAPB knockdown significantly reduced both cell-associated and supernatant virus titers. Moreover, VAPB depletion reduced cytoplasmic accumulation of virus particles and increased levels of nuclear encapsidated viral DNA. These results suggest that VAPB is an important player in the exit of primary enveloped HSV-1 virions from the nucleus. Importantly, VAPB knockdown did not alter pUL34, calnexin or GM-130 localization during infection, arguing against an indirect effect of VAPB on cellular vesicles and trafficking. Immunogold-labelling electron microscopy confirmed VAPB presence in nuclear membranes and moreover associated with primary enveloped HSV-1 particles. These data suggest that VAPB could be a cellular component of a complex that facilitates UL31/UL34/US3-mediated HSV-1 nuclear egress.

## 1. Introduction

Herpesvirus genome replication and encapsidation occurs in the host cell nucleus. The capsids at ~120 nm cannot pass through nuclear pore complexes [1], and so must traverse the nuclear envelope (NE) before virion maturation in the cytoplasm [2]. Capsids obtain a primary envelope from the inner nuclear membrane (INM), resulting in access to the NE lumen. They then exit the lumen by fusing with the outer nuclear membrane (ONM), leaving their primary envelope behind [3,4,5]. pUL31 and pUL34 are key viral proteins for HSV-1 nuclear egress. They recruit host kinases to locally dissolve the intermediate filament nuclear lamin polymer to allow nucleocapsid access to the INM [6,7,8,9] and it has been suggested that they may promote vesiculation in vitro and in vivo [10,11]. Though its specific function is not clear, HSV-1 pUS3 contributes to nuclear egress as primary enveloped particles accumulate in the NE lumen when it is depleted [12,13,14]. This is also observed for a combined knockdown of HSV-1 glycoproteins gB and gH that are also components of the final envelope [15,16]. However, as herpesviruses commonly co-opt host proteins [17,18] and cellular vesicle fusion requires many players [19,20], it has been suggested that host proteins might cooperate with viral proteins for efficient nuclear egress [21]. Several cellular proteins have been reported to be involved, including CD98 heavy chain, its binding partner β-integrin, and p32. CD98/β-integrin are fusion regulatory proteins and they accumulate at the nuclear membrane during HSV infection and interact with HSV-1 gB, gH, pUL31 and pUS3, all known to be involved in HSV de-envelopment [22]. p32 has been proposed to cause disassembly of the nuclear lamina by interacting with ICP34.5 [23]. This would indicate a role in primary envelopment but p32 can bind to HSV-1 pUL47, a major tegument protein, and through this interaction relocate to the NE, suggesting instead a role in de-envelopment [24]. 

It has been clearly established that transmembrane proteins can translocate between the nuclear membrane and the ER by a lateral diffusion mechanism [25,26,27,28]. Due to there being many connections between the ONM and the ER and between the ONM and the nuclear pores, a protein in the ONM could diffuse either to the ER or to the INM. Therefore, during HSV infection, any host proteins facilitating egress that fuse with the ONM might be expected to be found in both the NE and the ER, where they moreover might be observed to increase in abundance. Both INM proteins that become part of the primary envelope and ONM proteins supporting de-envelopment should have a reasonable likelihood of diffusing into the ER and so increasing in abundance there during an HSV-1 infection. To test this and to search for candidate viral fusion facilitators, these membrane components were prepared from mock versus HSV-1 infected cells and analyzed for proteins that both could be found in the NE in uninfected cells and increase in the ER in infected versus uninfected cells. A subset of vesicle fusion proteins (VFPs) were identified as possible fusion facilitators. We examined the role of the most-enriched VFP, vesicle-associated membrane protein-associated protein B (VAPB). VAPB depletion yielded reduced virus titers, and led to nuclear virion accumulation concomitant with a reduction in cytoplasmic virions. A subpopulation of VAPB located at the NE co-localized in part with pUL34. Nuclear virions, moreover, were stained for VAPB by immunogold-labelling electron microscopy. These data suggest that VAPB is an important contributor to the HSV-1 life cycle and may facilitate nuclear egress.

## 2. Materials and Methods

### 2.1. Cells and Viruses

HeLa and U20S cells were cultured in Dulbecco’s Modified Eagles Medium, 10% fetal calf serum, 50 µg/mL penicillin, 50 µg/mL streptomycin at 37 °C, 5% CO_2_. Wild-type HSV-1 was strain 17+. A modified strain carrying genomic replacement of VP26-RFP (kindly provided by Frazer Rixon, University of Glasgow, Glasgow, UK) was used for the experiment shown in Figure 1B, which visually determined the optimized time to take infected cells for protein preparation. For infections, the virus was adsorbed onto cells for 1 h at 37 °C. 

### 2.2. Preparation of Fractions

Nuclear envelopes (NEs) and microsomal membranes (MMs) were isolated from HeLa cells using established procedures [29]. For microsomes, we started with ~2 × 10^9^ cells from 15 roller bottle cultures. The post-nuclear supernatant was treated with 0.5 mM EDTA to inhibit metalloproteinases. Mitochondria and other debris were removed by pelleting 15 min at 10,000× *g*. Microsomes were floated through 1.86 M and 0.25 M sucrose layers by centrifugation in the SW28 rotor 4 h at 57,000× *g*, 4 °C. Material between the layers was diluted 4-fold with 0.25 M SHKM (50 mM HEPES pH 7.4, 25 mM KCl, 5 mM MgCl_2_, 1 mM DTT and 1.8 M sucrose) and pelleted at 152,000× *g* 1 h in a type 45 Ti rotor (Beckman, Brea, CA, USA). NEs were extracted with 0.1 N NaOH, 10 mM DTT, pelleted at 150,000× *g* for 30 min, and washed 3× in H_2_O. MMs were washed in H_2_O without NaOH extraction. The samples were divided for mass spectrometry and EM.

### 2.3. Mass Spectrometry

Pellets resuspended in 30 µL of 100 mM Tris-HCl pH 8.5, 8 M Urea were brought to 5 mM Tris(2-Carboxylethyl)-Phosphine Hydrochloride (TCEP) and incubated for 30 min RT. Reduction and alkylation used 10 mM chloroacetamide, 30 min in the dark. Endoproteinase Lys-C (Roche, Basel, Switzerland) was added at 0.1 mg/mL and incubated for 6 h, 37 °C. Following dilution to 2 M Urea with 100 mM Tris-HCl pH 8.5, 2 mM CaCl_2_, 0.1 mg/mL Trypsin, digestion was at 37 °C overnight. 5% formic acid quenched reactions and samples were centrifuged to remove undigested material.

The samples were analyzed by Multidimensional Protein Identification Technology (MudPIT) as previously described [30,31] with pressure-loading onto microcapillary columns packed with 3 cm of 5-μm Strong Cation Exchange (Luna; Phenomenex, Torrance, CA, USA), followed by 1 cm of 5 μm C18 reverse phase (Aqua; Phenomenex, Macclesfield, UK). These were connected to 100 µm columns pulled to a 5 μm tip containing 9 cm of reverse phase material. Peptides were separated on a Quaternary Agilent 1100 HPLC using a 10-step chromatography run over 20 h at 200–300 nL/min. Eluting peptides electrosprayed at 2.5 kV distal voltage into a LTQ linear ion trap mass spectrometer (Thermo Scientific, Waltham, MA, USA) with a custom-made nano-LC electrospray-ionization source. Full MS spectra were recorded on the peptides over 400 *m*/*z* to 1,600 *m*/*z*, followed by five tandem mass (MS/MS) events, sequentially generated in a data-dependent manner on the first to fifth most intense ions selected from the full MS spectrum (at 35% collision energy). Dynamic exclusion was enabled for 120 s. 

RAW files extracted into ms2 files [32] using RawDistiller v1.0 [33] were queried for peptide sequences using SEQUEST v.27 (rev.9) [34] against 55,691 human proteins (non-redundant NCBI 2014-02-04 release), plus 162 usual contaminants (e.g., keratin) and 77 NCBI RefSeq HSV-1 proteins. To estimate false discovery rates (FDRs), each non-redundant sequence supplemented the database after randomization, bringing the search space to 111,524 sequences. MS/MS spectra were searched without specifying differential modifications, but +57 Da were added statically to cysteines to account for carboxamidomethylation. No enzyme specificity was imposed and mass tolerance was set at 3 amu for precursor ions and ±0.5 amu for fragment ions.

Results from different runs were compared using DTASelect and CONTRAST [35] with criterion of DeltCn ≥ 0.08, XCorr ≥ 1.8 for singly-, 2.5 for doubly-, and 3.5 for triply-charged spectra, and a maximum Sp rank of 10. Merging peptide hits from all analyses established a master protein list (Appendix A). Identifications mapping to randomized peptides estimated FDRs that averaged 0.72% for proteins and 0.24% for peptides. Spectral-count based label free quantitation was used to estimate the relative levels of the proteins detected by mass spectrometry in each sample. In shotgun proteomics, the frequency of peptides being fragmented by the mass spectrometer (spectral counts) correlates with the abundance of the proteins these peptides derive from. Because longer proteins tend to generate more tryptic peptides, spectral counts are normalized by the protein molecular weight or length, defining a “spectral abundance factor” (SAF), which is further normalized against the sum of SAFs calculated for each protein/protein group detected in a sample. To deal with peptides shared between multiple proteins, distributed Normalized Abundance Factors (dNSAFs) are calculated for each non-redundant protein/protein group (Table 1, Table 2 and Appendix A), as described in [36]
$$dNSAF_{i}=\frac{dSAF_{i}}{\sum_{i=1}^{N}dSAF_{i}}\;\mathrm{with}\;dSAF_{i}=\frac{uSpC_{i}+\frac{uSpC_{i}\;}{\sum_{m=1}^{M}uSpC_{m}}\times sSpC_{i_{j}}\;}{Length_{i}}$$
in which shared spectral counts (sSpC) are distributed based on spectral counts unique to each protein *i* (uSpC), divided by the sum of all unique spectral counts for the M protein isoforms that shared peptide *j* with protein *i*. 

### 2.4. Bioinformatics

Ratios of HSV-1 infected MMs:mock-infected MMs and mock-infected NE:mock-infected MM were calculated from dNSAF values. Only proteins with an HSV-1 infected MMs:mock-infected MMs ratio higher than 1.3 and detected in the mock infected NEs were selected. Gene ontology (GO) analysis was performed in Ensembl BioMart. GO-terms and their corresponding child-terms were retrieved from the mySQL database http://amigo.geneontology.org [37]: “vesicle-mediated transport” (GO:0016192) “nucleocytoplasmic transport” (GO:0006913), “membrane organization” (GO:0061024), “regulation of protein phosphorylation” (GO:0001932). Relative proportions of these classes were calculated from the relative abundances of genes identified from each category, then compared to the proportions of all human-encoded proteins considering the same categories.

### 2.5. siRNA Knockdown of VFPs

HeLa cells were transfected for 48 h with 50 nM final concentration siRNAs to VAPB, Rab1a and Rab24 using JetPrime (Polyplus, New York, NY, USA). All siRNAs were Dharmacon (Lafayette, CO, USA) SMARTpools, except VAPB where a proven single siRNA oligo (Sigma, St. Louis, MO, USA) was used [38]. Control non-target siRNA was a scrambled firefly luciferase sequence (5′-CGUACGCGGAAUACUUCGA-3′). Following transfection, total protein was extracted for Western blot to determine knockdown or cells were infected with HSV-1 at MOI = 10 for the times indicated. 

### 2.6. Rescue Experiments

VAPB was cloned from keratinocyte cDNA into pCDNA3.1 and desensitized to the siRNA by changing CTTGGCTCTGGTGGTT to GTTTGCACTTGTCGTG using CloneAmp HiFi PCR Premix (Clontech, Mountain View, CA, USA). Mutations were verified by Sanger sequencing. Other siRNAs targeted 3′ untranslated regions, so mutagenesis was not required for their rescue constructs. HeLa cells expressing rescue constructs were transfected with siRNA for 48 h, then infected with HSV-1 at MOI = 10 for 16 h. As indicated in the figure legends, either released virus was collected by pelleting from the supernatant or cell-associated virus was collected after extensive washing followed by pelleting cells. Titrations were carried out on U2OS cells as described [39]. U2OS cells were used for titrations because they are the most optimized for virus entry and replication and thus enable better measurement of the number of infectious particles. 

### 2.7. Isolation of Nuclei and DNA Preparation for qPCR

Cells were scraped into 1 mL PBS then pelleted by centrifugation in an Eppendorf microfuge 1 min at 8000× *g*, 4 °C and resuspended in 400 µL hypotonic Buffer A (10 mM Hepes, 1.5 mM MgCl_2_, 10 mM KCl, 0.5 mM DTT, pH 7.9). Cells were lysed by 15 syringe passages through a 21-gauge needle following 15 min incubation on ice. The nuclei were pelleted by centrifugation then broken by sonication. Nuclei were resuspended in DNase digestion buffer and incubated for 1 h, 37 °C using 10 units RQ DNAse 1 (Promega, Madison, WI, USA) followed by enzyme inactivation. DNase-resistant nuclear DNA was prepared using a QiAamp DNA mini kit (Qiagen, Germantown, MD, USA) incorporating RNase and proteinase K digestion. 

### 2.8. qPCR

siRNA-transfected cells were infected with HSV-1 WT and supernatants or pellets were collected at 16 hpi. Viral DNA was purified as above. qPCR was carried out on an Applied Biosystem 7500 machine with Takyon^TM^ Low Rox Probe MasterMix dTTP Blue (Eurogentec, Liege, Belgium) using primers detecting gD HSV-1 sequence [40] (Forward: 5′-CGGCCGTGTGACACTATCG-3′, Reverse:5′-CTCGTAAAATGGCCCCTCC-3′, Probe: FAM-CCATACCGACCACACCGACGAACC-TAMRA) and a cellular GAPDH control (Forward: 5′-CGCTCTCTGCTCCTCCTGTT-3′, Reverse:5′-CCATGGTGTCTGAGCGATGT-3′, Probe: 5′-CAAGCTTCCCGTTCTCAGCC-3′). Serial dilutions of known-titer HSV-1 were used to create a standard curve in genomes/mL. 

### 2.9. Western Blotting

Proteins were separated by SDS-PAGE, transferred to nitrocellulose membranes, and blocked with 5% milk powder in PBS containing 0.1% Triton-X-100 (PBST) for 1 h, RT. Primary antibodies were applied for 1–2 h, RT or overnight, 4 °C, washed in PBST, then incubated with anti-mouse IgG or anti-rabbit IgG conjugated to horseradish peroxidase for 1 h. Following washing in PBST, membranes were visualized either using ECL (GE Healthcare) and exposed to Kodak X-Omat S film or analyzed directly on a LICOR Odyssey imager (LI-COR Biosciences, Lincoln, Ne, USA) using antibodies conjugated to fluorescent markers. Antibodies: GM-130 (EP892Y, Abcam, Cambridge, UK), GAPDH (E1C604-1, Enogene, Calnexin (SPA-860, Stressgen, San Diego, CA, USA), lamin A and C (3262, 3931), Rab1a (Santa Cruz Biotechnology, Dallas, TX, USA), Rab24 (BD bioscience, San Jose, CA, USA), gC (ab6509, Abcam, Cambridge, UK), VAPB mouse monoclonal antibody (66191-1, Proteintech, Manchester, UK), and US3 (rabbit polyclonal) and UL34 (rabbit polyclonal) antibodies were kindly provided by Dr. Thomas Mettenleiter and Dr. Barbara Klupp (Friedrich-Loeffler-Institut, Greifswald, Germany). Dr. Christopher C.J. Miller (Kings College London, London, UK) kindly provided the VAPB rabbit antibody.

### 2.10. Fluorescence In Situ Hybridization (FISH)

HeLa cells cultured on coverslips were washed in PBS prior to fixation in 4% para-formaldehyde, 1× PBS for 10 min at RT. Cells were permeabilized 6 min with 0.2% Triton-X-100 in PBS then pre-equilibrated in 2× SCC and treated with RNase A (100 µg/mL) at 37 °C, 1 h. Following washing in 2× SCC, cells on coverslips were dehydrated with a 70%, 85%, 100% ethanol series, then air dried, heated to 70 °C and submerged into 85 °C preheated 70% formamide, 2× SSC (pH 7.0) for 18 min. A second ethanol dehydration series was performed. 150–300 ng biotin-labelled probe was added in hybridization buffer (50% formamide, 2× SSC, 1% Tween20, 10% Dextran Sulphate) containing 6 µg human Cot1 DNA (Invitrogen, Waltham, MA, USA) and sheared salmon sperm DNA and incubated at 37 °C, 24 h. An HSV-1 ICP27 plasmid was end-labelled. After incubation with this probe, coverslips were washed 4 × 5 min in 4× SSC, 50 °C, followed by 4 × 5 min washes in 0.1× SSC, 65 °C, then pre-equilibrated in 4× SSC, 0.1% Tween-20 and blocked with 4% BSA before incubating 30 min, RT with Alexa Fluor^®^ (Jackson ImmunoResearch Laboratories, West Grove, PA, USA) conjugated-Steptavidin antibodies and 4,6-diamidino-2 phenylindole, dihydrochloride (DAPI) at 2 µg/mL. Coverslips were washed 3 times in 4× SSC, 0.1% Tween-20, 37 °C and mounted on slides in Vectashield (Vector Labs).

### 2.11. Fluorescence Microscopy

HSV-1 or mock-infected HeLa cells grown on coverslips were fixed 10 min in 3.7% formaldehyde in PBS, then permeabilized for 5 min, RT with 0.2% Triton X-100. Blocking was with PBS containing 10% human serum, 10% donkey serum and 1% BSA,1 h, RT. Primary antibodies were applied for 1 h at RT: VAPB (66191-1, mouse monoclonal, Proteintech, Manchester, UK; 1:200), US3 (rabbit polyclonal; 1:500) and pUL34 (rabbit polyclonal; 1:500); both rabbit antibodies were kindly provided by Thomas Mettenleiter and Barbara Klupp (Friedrich-Loeffler-Institut, Greifswald, Germany). Following 1 h incubation with Alexa-fluor secondary antibodies (Jackson ImmunoResearch Laboratories, West Grove, PA, USA; 1:1000) and 4′,6-diamidino-2-phenylindole (DAPI; 1:2000) in blocking buffer and washing, the coverslips were mounted with Vectashield. All secondary antibodies were donkey minimal cross-reactive conjugates to Alexa-fluor dyes matched to the Sedat quad filters used (488, 568, and 647) to prevent signal bleedthrough and negative controls were generally done to further confirm lack of cross-reactivity. Images were acquired on a Nikon TE-2000 microscope using a 1.45 NA 100× objective, Sedat quad filter set, PIFOC Z-axis focus drive (Physik Instruments, Cranfield, UK), and a CoolSnapHQ High Speed Monochrome CCD camera (Photometrics, Tucson, AZ, USA) run by Metamorph software. This provided a pixel size of 0.0645 µm^2^. For general widefield images shown in Figures 1B, 3B, 6B and 7A, an image was taken from a focus point at the midplane of the nucleus as this generally excludes signal from any other staining above or below the nucleus and generally affords the widest view of the ER as well outside the nucleus. For Figure 3E images, image stacks (0.2 μm steps) were deconvolved using AutoquantX (Media Cybernetics, Rockville, MD, USA). Images in Figure 6A were taken using a Zeiss LSM510 Meta confocal microscope. Micrographs were saved from source programs as 12-bit.tif files and analyzed with Image Pro Plus software and/or prepared for figures using Photoshop CS6. Images stained with the same antibodies were taken using identical settings and exposure times. For FISH experiments, DAPI images were used as a mask in ImageJ to determine nuclear area, and the total fluorescence intensity in the nucleus versus the whole cell was determined for >100 cells for each condition. For NE:ER ratio quantification, original images were analyzed in ImageJ using the edge of a DAPI image mask and taking 5 pixels in either direction to define NE fluorescence intensity. Given the 100–200 nm limit of resolution for standard (non-super resolution) fluorescence microscopy and the architecture of the NE, this 300 nm distance from the edge of the DNA signal should capture the full NE signal. ER fluorescence intensity was determined from pixels outside the NE ring and mean pixel intensities were determined by dividing total intensities by area. The NE/ER value for >50 cells per condition was determined and means plotted. 

### 2.12. Electron Microscopy

MM pellets were washed in sterile H_2_O and fixed with 2.5% glutaraldehyde and 1% osmium tetroxide, then dehydrated through a graded alcohol series and embedded in Epon 812. siRNA-treated and HSV-infected HeLa cells were fixed with 2.5% glutaraldehyde in 2% sucrose, 0.05M Cacodylate buffer, pH 7.2 overnight at 4 °C. Cells pelleted by centrifugation were fixed with 1% osmium tetroxide (TAAB Labs, Aldermaston, UK) and stained for 1 h, RT with 2% aqueous uranyl acetate. Cells were pelleted through 3% low melting temperature agarose (Geneflow, Lichfield, UK) at 45 °C. The agarose was set at 4 °C and cell pellets were cut into ~1 mm cubes, then dehydrated through a graded alcohol series (30–100%) and embedded in Epon 812 resin (TAAB Labs, Aldermaston, UK) followed by polymerization for 3 days at 65 °C. 120 nm sections were cut with a UC6 ultramicrotome (Leica Microsystems, Wetzlar, Germany) and examined in a JEOL 1200 EX II electron microscope, recording images on a Gatan Orius CCD camera. Immunoelectron microscopy was performed using the Tokuyasu method [41]. For imaging, a Hitachi H7600 TEM was used at 100 kV. 

### 2.13. Data Availability

The mass spectrometry dataset [raw, peak, search, and DTASelect result files) can be obtained from the MassIVE database via MSV000079886, from the ProteomeXchange via PXD004519, and from the Stowers Original Data Repository at LIBPB-1083. 

## 3. Results

### 3.1. Identification of Cellular Proteins Potentially Involved in Viral Egress 

Cellular proteins involved in nuclear egress must be present and might even accumulate in the NE during infection. Moreover, if they become part of the primary envelope and/or are involved in de-envelopment fusion with the ONM, they may specifically accumulate in the ONM as nuclear egress progresses and possibly also in the contiguous ER after fusion with the ONM. Although their residence in both membranes could be brief due to recycling back to the INM, we reasoned that a proteomics analysis of NE and ER membrane fractions may provide some clues as to the types of cellular proteins that might take part in HSV nuclear egress. We analyzed HeLa nuclear and ER (microsomal) membranes around the time of viral nuclear egress but before secondary envelopment. pUS3, a serine/threonine viral kinase that contributes to primary envelope fusion with the ONM, is expressed before primary envelopment/de-envelopment and peaks roughly when secondary enveloped particles are first detected [13,14]. By contrast, surface glycoprotein gC is expressed later in infection. Western blotting showed that pUS3 began to be highly expressed at 9 h post-infection (hpi), reaching maximal levels by 12 hpi (Figure 1A). At this time, gC was only becoming detectable. A second analysis found that in infected HeLa cells, RFP-labeled capsid protein VP26 was in punctate intranuclear spots at 8 hpi, but had accumulated in larger spots at the NE by 9 hpi that persisted at 10 hpi (Figure 1B). Therefore, the isolation of NE and microsomal membrane (MM) fractions was performed between 8 and 9 hpi from mock-infected and HSV-1 infected (MOI = 10) HeLa cells. The fractions displayed a number of common and unique components (Figure 1C). Western blotting revealed that Lamin A/C and LAP2β were only found in the NE fraction as expected, while ER marker calnexin was predominantly in the MM fraction with some in the NE due to ONM-ER continuity (Figure 1D). Neither NE nor MM fractions contained the Golgi marker GM-130, suggesting low contamination with Golgi membranes involved in secondary envelopment (Figure 1E). Electron microscopy of the isolated HSV-1 MMs revealed the expected membrane vesicles. Enveloped virus particles were also captured in some vesicles as expected due to the contiguous nature of the NE and ER (Figure 1F).

### 3.2. Host and Viral Proteins Identified in HSV-1 Infected MMs

MMs from HSV-1 infected cells and NEs and MMs from mock-infected cells were analyzed by mass spectrometry using Multi-dimensional Protein Identification Technology (MudPIT) LC/LC/MS/MS [30,31]. Due to the presence of virions (Figure 1F), HSV-1 proteins were identified in the infected MMs. All but pUL17 were detected out of 8 viral proteins previously found in a study of primary enveloped virions [42]. We identified pUL31 and its partner pUL34 (Table 1, Appendix A), key players in primary envelopment. Other identified tegument proteins and glycoproteins included gD, gB, gC all reported to associate with HSV-1 primary virions in different studies [43,44] with gB being specifically implicated in ONM fusion during egress [15,16].

Gene ontology (GO) analysis of the proteins identified revealed many proteins with functions in vesicle-mediated transport. Although this category reflects <4% of genes encoded in the human genome, it represented 10% of all proteins detected in the MMs and 12% of all proteins detected in the HSV-1 infected MMs. Moreover, when increasing selection stringency to look at only proteins enriched in the HSV-1 infected MMs, the category jumped to 20% of total proteins. Because it also makes sense scientifically that VFPs could have a potential role in HSV nuclear egress, we chose to pursue this category. We increased selection stringency, only considering VFPs with at least 5 spectra in the NE fraction to suggest that they are reasonably abundant in the NE before infection. We then plotted the abundance ratio of these proteins in the HSV-infected:mock-infected MMs using dNSAF values, a measure of abundance based on both spectral counts and percentage of total mass in the protein fraction [36] (Figure 2A, Table 2). The ratios of NE:mock-infected MM dNSAF values were also plotted (Figure 2B). It is noteworthy that, while the relative ratios are relevant here, it is not possible to compare absolute amounts in the NE and MMs because of the differences in their volumes and because there are much greater losses of membranes when isolating MMs. Though many VFPs identified would potentially make sense to contribute to nuclear egress, VAPB was selected as a candidate for further analysis because it was enriched in the NE compared to MMs, the most enriched VFP in HSV-1 infected versus mock-infected MMs, and recently indicated to interact with the inner nuclear membrane protein emerin in the IntAct Molecular Interaction Database “https://www.ebi.ac.uk/intact/”. 

### 3.3. VAPB Expression and Localization During HSV-1 Infection

VAPB is a type II integral membrane protein previously characterized in the ER. Together with family member VAPA, it is thought to function with cytoplasmic vesicle transport proteins and cytoskeletal elements to maintain membrane structure and facilitate lipid transport, membrane trafficking and membrane fusion [45]. VAPB is a C-terminally anchored protein with its primary mass facing the cytoplasm in the ER (nucleoplasm for the inner nuclear membrane population) [46]. The N-terminal region has a major sperm protein (MSP) homology domain that interacts with FFAT motif (two phenylalanines in an acidic track) proteins [47] followed by a coiled-coil domain before the transmembrane segment (Figure 2C). A mutation (P56S) has been reported in the VAPB MSP domain causing an autosomal dominant form of amyotrophic lateral sclerosis (ALS8) that results in VAPB aggregation and neurotoxicity [48].

VAPB-increased abundance in the infected MMs could be due to HSV-1-induced upregulation of protein expression, virus-induced ER recruitment, or to increased abundance in the membranes due to HSV-1 envelopment/de-envelopment. Total VAPB protein levels in the cell were constant during a time course of infection (Figure 2D). VAPB has not previously been directly tested for a function in the NE, but its initially reported restriction to the ER has been challenged [46,49,50,51,52]. To confirm a NE location, a very specific monoclonal antibody was used (Figure 3A). In the mock-infected cells, considerable ER distribution of VAPB was observed as well as a weak, but distinct rim around the nucleus (defined by the DNA staining) and this staining was largely similar in infected cells (Figure 3B). To identify infected cells, mock and HSV-1 infected cells were co-stained for VAPB and the herpesvirus US3 protein. Next, an algorithm was employed that measured NE versus ER fluorescence and the mean VAPB fluorescence signal intensities were calculated to identify any redistribution of VAPB (Figure 3C). A small, but statistically significant increase in the NE pool was observed over the early stages of infection. Over 50 cells were quantified for each condition and no difference was observed in the NE:ER ratio of calnexin during HSV-1 infection. It is important to note, however, that the increase in the NE covered a wide distribution, which might reflect different cells at different stages of infection or might reflect shuttling of VAPB if it is being recycled for repeated use at the NE. Nonetheless, even with the wide distribution, the data indicated the NE:ER ratio increasing from 1.18 to 1.80 between the mock and 16 hpi conditions, respectively (Figure 3D). Finally, we tested for co-localization between VAPB and NE-located pUL34. The majority of VAPB signal was in the ER while pUL34 concentrated in the nucleus, but both gave clear nuclear rim staining in addition to staining in the ER with the NE pool exhibiting partial overlap between the signals (Figure 3E). Lower left panels are blowups from the above images, better showing the concentration of VAPB and pUL34 at the NE and that there are both areas with a complete distinction between VAPB (red) and pUL34 (green) signals and yellow areas where the signal overlaps. To quantify distribution, in the middle graph, each signal was quantified in the NE as defined by the perimeter of the DNA staining and signal for the rest of the cell. For both proteins, more signal was observed in the NE than in the rest of the cell. To quantify co-localization, we used Pearson’s Correlation Coefficient to quantify overlap VAPB with pUL34 in the NE or the rest of the cell. A much stronger correlation was observed for the NE signal than that in the rest of the cell. The data suggest a partial recruitment of VAPB to the NE during HSV-1 infection.

### 3.4. Knockdown of VAPB Yields Significant Reduction of HSV-1 Viral Titers

To test the role of VAPB in virus replication, siRNA depletion followed by HSV-1 infection was carried out. Rab24, a regulator of intracellular trafficking, was used as a negative control because it did not increase in HSV-1 infected MMs compared to mock-infected MMs. Rab1a was used as a positive control because this protein, required for ER-to Golgi complex transport [53], is involved in HSV-1 mature particle assembly (secondary envelopment) and its knockdown reduces viral growth by 60% [54]. Western blotting showed that VAPB was depleted to nearly undetectable levels and both controls were knocked down by roughly 80% (Figure 4A). As expected [55], Rab24 knockdown had little effect on viral titers while Rab1a knockdown reduced cell-released viral titers by 62%: VAPB exhibited an even stronger reduction in cell-released virus titers of >90% (Figure 4B). These were specific effects of the siRNAs as rescue experiments yielded nearly full recovery of virus titers (Figure 4C). Similar to Rab1a, VAPB depletion resulted in a marked reduction in cell-associated virus titers (<80%) compared to control Rab24 (Figure 4D); yet, viral genomes were still abundantly produced (Figure 4E). Moreover, these effects were not likely due to inhibition of production of viral proteins as pUL34 and gC levels were not reduced in VAPB knockdown cells (Figure 4F). A multistep growth curve (starting MOI = 0.1) revealed around a two log-fold difference between HSV-1 replication in mock versus VAPB siRNA-treated cells (Figure 4G).

### 3.5. VAPB Knockdown Yields Reduced Cytoplasmic Virus Particles and Nuclear Particle Accumulation

Viral titer reduction could reflect a function of VAPB in nuclear egress or secondary envelopment or cytoplasmic transport. Examination by electron microscopy of HSV-infected, and control- or Rab24-depleted cells revealed the expected distribution with many virus particles present both in the nucleus and in the cytoplasm at 16 hpi (Figure 5A, upper panels). However, VAPB knockdown resulted in more nuclear and less cytoplasmic virus particles compared to control siRNA-treated cells (Figure 5A, lower panels). Images focused on the cytoplasm revealed far fewer cytoplasmic particles in the VAPB-depleted cells (Figure 5B). Quantification of particles in the nucleoplasm, the NE and the cytoplasm revealed an increase in nucleoplasmic and NE virions with a corresponding reduction in cytoplasmic virions in VAPB knockdown cells (Figure 5C,D). Several images revealed enveloped particles in lumenal extensions of the NE and the presence of the virus envelope could clearly be distinguished by size from the un-enveloped particles in the nucleoplasm (Figure 5A, lower right panel). Note that the NE lumen is similarly enlarged by accumulating virus particles in US3 deletions [12,13].

Next, we tested for capsid protein VP5, which in a normal infection generally appears predominantly at the nuclear periphery and aggregates in the cytoplasm because of the rapid trafficking of assembled virions from the nucleus [56]. This is the pattern observed for control siRNA cells, while the VAPB knockdown cells show a strong accumulation of the VP5 in the nucleus (Figure 6A). We also used fluorescence in situ hybridization (FISH) and viral DNA qPCR to quantify viral genomes. Infected cells with control and VAPB knockdowns were fixed and hybridized with a probe for the HSV-1 ICP27 gene. Pre-treatment with RNase ensured only viral genomes would be recognized. A proportion of nuclear virus genomes will not be encapsidated and this approach will also detect them; however, the aim was to determine egress to the cytoplasm by measuring fluorescent viral genome signals in both the nucleus and cytoplasm. Co-staining with DAPI identified the nuclear boundaries and imaging revealed virus accumulating in the cytoplasm in non-target siRNA control cells, the Rab24 knockdown, and in cells depleted of Rab1a that affects HSV-1 egress in the cytoplasm (Figure 6B). By contrast, in the VAPB knockdown, the FISH signal for viral genomes was visually restricted to the nucleus (Figure 6B). Images for this analysis were taken from the nuclear midplane where nuclear area is greatest. While generally the ER sectional area is also greater at the nuclear midplane, it is possible that changes induced by the HSV-1 infection that alter both ER and Golgi membrane distribution might yield a skewed distribution for the positioning of virus particles. Nonetheless, all samples were treated equally and so the differences between the VAPB knockdown and control samples should still be relevant. The intensity of total FISH signal in each cell, and also that just in the nucleus (using the DAPI staining as a mask) was determined. The total signal divided by the nuclear signal was plotted so that the amount of signal outside the nucleus is reflected in values above 1 (Figure 6C). All three controls exhibited a clear cytoplasmic signal while VAPB knockdown had a value close to 1, indicating mainly nuclear signal. Interestingly, some accumulation of virus genomes at the NE could be observed in the VAPB knock-down (Figure 6D). Next we quantified DNase-resistant viral DNA (encapsidated genomes) in the nucleus of HSV-infected cells with or without siRNA depletion. Figure 6E shows a significant increase in DNase-resistant viral genome copies (4.5-fold) when VAPB was depleted compared to control Rab24. Rab1a depletion resulted in more DNase-resistant nuclear genome copies than the control, but this difference was not statistically significant. 

### 3.6. VAPB Knockdown Does Not Interfere with Virus Protein Accumulation at the NE

VAPB knockdown was previously reported to interfere with NE targeting of NPC proteins gp210 and Nup214 and the INM protein emerin [38]; thus, its knockdown could potentially interfere with INM accumulation of pUL34 that is essential for primary envelopment. To test if VAPB indirectly alters egress, we stained for pUL34 in HSV-1 infected VAPB-depleted cells (Figure 7A). No notable difference was observed in pUL34 distribution between control (siCntl) and the VAPB knockdowns. NE rim staining for pUL34 was observed in individual cells with strong VAPB knockdown. These data show that VAPB contributions to nuclear egress are not indirect effects on the distribution of the most critical viral proteins involved in this step.

Confocal and EM images did not indicate major changes in the NE upon VAPB depletion. In case VAPB depletion resulted in gross changes to other cellular membranes that might affect HSV cellular egress, we determined the location of calnexin, an ER marker, and GM-130, a Golgi marker (Figure 7B). The pattern of staining with antibodies against either protein was generally similar between control and VAPB depleted cells, suggesting that VAPB depletion does not affect the gross structure of other membranous compartments though minor effects on Golgi structure and function remain possible.

### 3.7. VAPB Is Present on the NE and in Association with Viral Particles

Immunogold labelling electron microscopy was used to visualize VAPB in the NE. Although typically only 3–5 particles were observed in a NE region captured in a particular section, nearly all images examined had particles in both INM and ONM in both the mock-infected and HSV-1 infected cells (Figure 8A,B). As expected, particles were also observed in association with the ER (Figure 8A, ER membrane delineated by arrowheads). Importantly, some virus particles inside nuclear invaginations that appear to be luminal extrusions contained VAPB-labelled gold particles (Figure 8C). These generally displayed tightly packaged genomes, an apparent envelope, and were proximal to other cellular membranes and DNA, suggesting that they are primary enveloped particles captured in the NE lumen. The image in Figure 8D shows a particle that is clearly in the NE lumen (L). Other sections that appear to be glancing sections at the nuclear surface often contained many virus particles with less densely packaged genomes associating with VAPB-labelled gold particles (Figure 8E). The staining for VAPB on nuclear virus particles suggests the possibility that VAPB participates in the process of primary envelopment.

## 4. Discussion

It has been proposed that host cell proteins could play a role in primary envelopment and nuclear egress [21] and such a role for several host proteins has already been indicated. During HSV-1 infection activated protein, kinase C is recruited to the nuclear membrane, where it phosphorylates and remodels the nuclear lamina to facilitate primary envelopment [57]. p32 is a cellular protein that can interact with a number of herpesvirus proteins and in HSV-1 infection, it is recruited to the nucleus/nuclear rim by ICP27 and ICP34.5 [23,58], but may play a role in nuclear egress via interaction with tegument protein UL47 [24]. Two proteins, CD98 and β-integrin, can act as regulatory proteins in the case of fusion of enveloped viruses with the plasma membrane [59,60,61,62] but during HSV-1 infection, they can also be recruited to the nuclear membrane where, through interaction with key viral proteins including UL31 and US3, they may indirectly control de-envelopment [22].

Our approach to identifying additional cellular proteins was to examine changes in the proteome of NE and ER membranes comparing HSV-1 infected and mock infected HeLa cells. While there are many proteins that might change in levels due to HSV-1 infection or relocate to these membrane compartments for other reasons as a consequence of HSV-1 infection, it is likely that within the set of changing proteins would also be those involved in egress. This is because participation in egress requires first a physical presence in the NE and any INM proteins captured in the primary envelope would enter the ONM upon fusion, from whence they could diffuse into the ER. Thus, while proteins fulfilling these criteria would not necessarily be involved in egress, it is likely that a subset would. Therefore, this approach was used to identify candidates to be further tested for possible functions in egress. Interestingly, VFPs comprised of a key protein group whose presence in the NE/ER was significantly increased during HSV-1 infection. In particular, among roughly 50 VFPs identified in the NE, only a few exhibited the expected characteristics, thus identifying as a candidate VAPB, an ER resident protein whose levels do not change during HSV infection and which we found to be largely required for HSV-1 replication. It facilitates cytoplasmic viral particle accumulation, localizes to the NE in association with primary enveloped virions and co-locates with viral nuclear egress protein pUL34. These data suggest a role for VAPB in HSV-1 nuclear egress. Although exogenous overexpression of viral pUL31/34 proteins promotes formation of NE vesicles and their in vitro expression can bend membrane vesicles [10,11], it seems likely that during infection vesicle fusion, proteins could be used to make egress more efficient. Herpesviruses frequently co-opt host cell machinery to support various aspects of their life cycle [17,18] and ESCRT pathway proteins were recently shown to be involved in nuclear egress of Epstein-Barr virus (EBV) [63]. In this case, the Chmp4b protein important for scission complex assembly co-localized in perinuclear aggregates with EBV protein BFRF1 and inhibition of the Alix bridging protein yielded capsid protein accumulation in the nucleus [63]. Notably these proteins were not required for HSV-1 egress, suggesting that other host cell vesicle trafficking proteins function instead of the ESCRTs for this virus.

VAPB is a member of the Vesicle-associated membrane protein (VAMP)-Associated Protein family of C-terminal tail anchored proteins. It has been shown to function as an adaptor to recruit target proteins to the ER and execute cellular functions such as lipid transport, membrane trafficking and membrane fusion [45]. It also functions in calcium homeostasis and has an additional mitochondrial function reported [49,52]. An indirect NE link was reported for VAPB, where its knockdown resulted in reduced NE accumulation of NPC proteins gp210 and Nup214 and the INM protein emerin [38]. However, it was thought that this reflected an indirect consequence of VAPB knockdown, inducing Golgi fragmentation and disruption of ER to ERGIC (ER-Golgi Intermediate Compartment) to Golgi trafficking [64,65]. As we found clear VAPB accumulation in the NE, these data might be re-interpreted to consider a nuclear membrane trafficking role and/or a role in fusion events during NE assembly at the end of mitosis. The latter is consistent with the ESCRT proteins shown to be involved in EBV nuclear egress also functioning in NE reformation after mitosis [66]. Our data are also consistent with a potential direct function in nuclear egress as VAPB knockdown did not block NE accumulation of pUL34. This does not discount, however, the possibility of VAPB having an additional role in secondary envelopment. 

A critical outstanding question is whether VAPB contributes to primary envelopment or de-envelopment or both during nuclear egress. Immunogold EM labelling revealed VAPB in association with nucleoplasmic/luminal virions in a membranous vesicle and, together with the nucleoplasmic accumulation of virus particles, this suggests a role in primary envelopment. As VAPB is in both the INM and ONM according to the immunogold EM data, it could potentially contribute to both primary envelopment and de-envelopment; however, the topology of VAPB is such that its principal functional mass should be facing the nucleoplasm in the INM and the cytoplasm in the ONM. Thus, it is perfectly poised to potentially interact with virus nucleocapsids during primary envelopment, but it should not be positioned to interact with the primary enveloped particles in the nuclear envelope lumen in order to initiate fusion with the ONM for de-envelopment unless, as has never been tested, it assumes a different topology during HSV-1 infection. Nonetheless, once the initial fusion has begun, VAPB either in the primary envelope or in the ONM could potentially be positioned to contribute to stabilizing and/or completing the process of nuclear egress. Likewise, and perhaps more likely, the separate role reported for VAPB in calcium homeostasis could lead to indirect effects on the fusion step [49,52]. In considering a potential de-envelopment function for VAPB, it is noteworthy that previous studies of HSV-1 US3, gB and gH depletion or mutants revealed an accumulation of primary enveloped particles in both the nucleoplasm and in the NE lumen, the expected phenotype for a role in the de-envelopment phase of nuclear egress [12,13,15,16] and, similarly, both nucleoplasmic and luminal viral particle accumulation were observed with VAPB knockdown. However, further experiments are required to identify the precise stage at which VAPB acts, and these are ongoing.

A potential function for VAPB in nuclear egress does not contradict recent papers arguing that pUL31 and pUL34 are sufficient for primary envelopment, as these studies showed induction of membrane invaginations in vitro in one study [10] and in context of a cell still containing VAPB in the other [11], while not addressing the issue of efficiency compared to a wild-type infection. Though with greatly reduced titers, HSV-1 has been shown to still get out of the nucleus in the absence of pUL31/34 [12,13,14,67,68]. Nonetheless, as pUL31/34 are central proteins for nuclear egress, it will be important to determine if VAPB contributions to nuclear egress result from participating in functions with or independent of these critical viral proteins. Thus far, our efforts to test for an interaction between VAPB and pUL34 have been negative; however, VAPB could also interact with HSV-1 capsid/tegument proteins or contribute in as yet undetermined ways. The lack of clarity in the composition of primary enveloped particles allows for a very wide range of potential partners, so this may be a significant undertaking in the future. If VAPB interacts with virus proteins, it could also provide a mechanism for specific recruitment of VAPB to egress complexes; however, it is also possible that HSV-1 disruption of ER and Golgi membranes unintentionally results in a redistribution of VAPB where it can indirectly contribute to nuclear egress and it remains possible that the trafficking of some important player in nuclear egress besides pUL34 is not able to gain access to the NE in the VAPB knockdown cells.

The demonstration of significant defects in virus nuclear egress for the VAPB knockdown suggests that other of the VFPs similarly highlighted by our study might also be involved in HSV-1 nuclear egress. This idea is more compelling by the general functioning of VFPs in larger complexes and observations that VAPB itself interacts with other proteins identified or closely related proteins. For example, Rab11, which was also highlighted by our mass spectrometery analysis, functions in a complex with VAPB and several other proteins [69,70]. Separately VAPB has been shown to interact with VAMP1 and 2 [51], both members of the same family, as VAMP7 that was similarly highlighted in our mass spectrometry analysis. Moreover, the IntAct Molecular Interaction Database [71] reveals that VAMP7 is involved in several different vesicle membrane trafficking complexes including multiple SNARE complexes, one of which includes Stx7 that was also highlighted by our mass spectrometry analysis. Compellingly, IntAct also reveals VAPB interactions with the inner nuclear membrane protein emerin and multiple virus proteins from Hepatitis C Virus [72].

A study reporting co-localization of gB, gH, pUL31 and pUL34 at the NE together with cellular proteins CD98hc and β1-intergrin argued for formation of a complex with a role in HSV nuclear de-envelopment [22]. Just as the viral proteins in that study might co-opt host cellular proteins, we postulate that VAPB forms part of a larger egress complex together with viral proteins [73]. The reduction in cytoplasmic particles and virus titers with VAPB knockdown and the labelling of virus particles with VAPB antibodies strongly argues that VAPB is directly involved in HSV-1 nuclear egress. 

## Figures and Tables

**Figure 1 cells-08-00120-f001:**
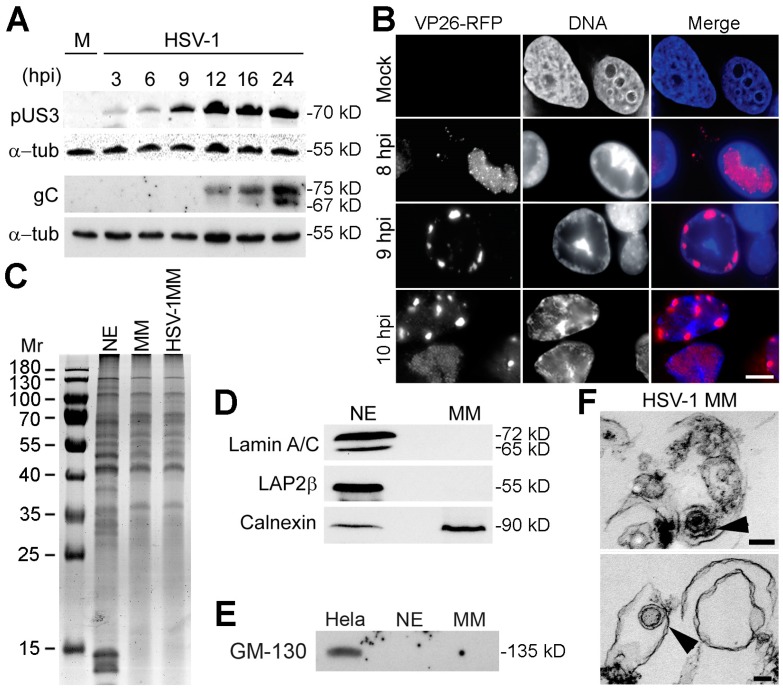
Optimization of conditions to isolate HSV-1 infected MMs. (**A**) An expression timecourse of US3 and gC viral proteins was analyzed by Western blot to determine the optimal time to isolate HSV-1 infected MMs before significant secondary envelopment had occurred. Cell lysates from Hela cells infected at MOI = 10 were prepared at indicated times post-infection. α-tubulin was used as a loading control. (**B**) Fluorescence microscopy images of an HSV-1 strain with VP26 capsid protein fused with RFP in Hela cells after 8, 9 and 10 hpi. At 8 hpi, most of the VP26 signal was located inside the nucleus of infected cells. At 9 hpi, cells started to show perinuclear localization of VP26. Scale bar, 10 µm. (**C**) Coomassie-stained gel of mock-infected NE, mock-infected MM, and HSV-1 infected MM fractions showed a clear difference in protein composition between NEs and MMs, but no notable visible differences between infected and mock-infected MMs. (**D**) Western blot of fractions from (**C**) stained with ER and NE markers to determine fraction purity. Infected MMs are shown. The ER marker calnexin was present in both NE and MM fractions as expected, because the ONM is continuous with the ER and many proteins are shared. In contrast, the NE markers lamin A/C and Lap2β were absent from MMs. Similar amounts of total protein were loaded. (**E**) To test for possible Golgi contamination in the preps, the NE and MM fractions were blotted for Golgi marker GM-130. Equal protein loading from HeLa cells confirmed that the antibody was working. (**F**) Ultrastructure of isolated MMs from HSV-1 infected HeLa cells. Electron micrographs showed the characteristic single-membrane structure of the MMs. Arrows point to electron-dense symmetrical structures of around 100 nm diameter that probably represent primary viral particles with the bottom one likely lacking a packaged genome. Scale bars, 100 nm.

**Figure 2 cells-08-00120-f002:**
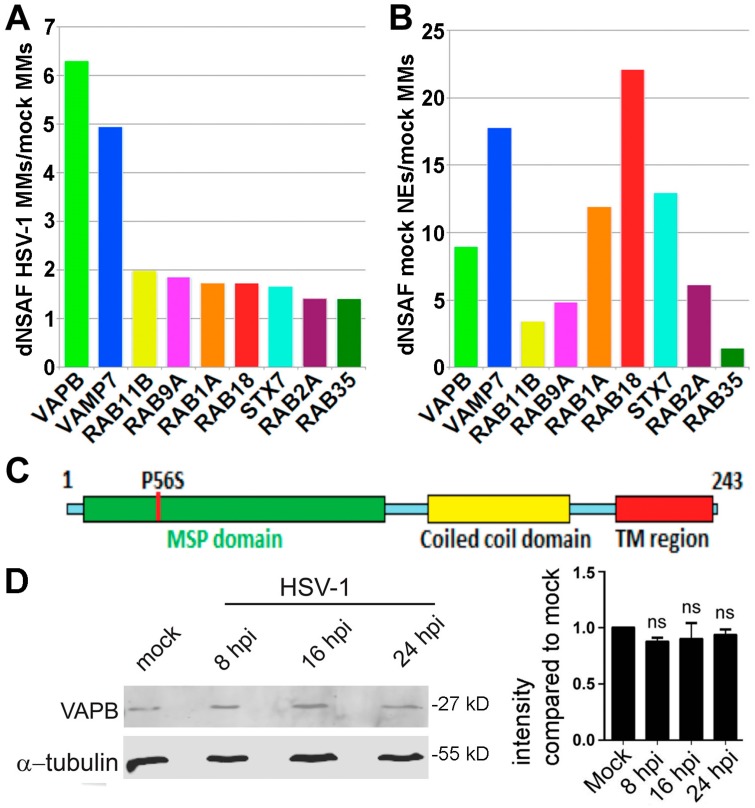
Enrichment of VFPs in HSV-1 infected MMs identified by mass spectrometry. (**A**,**B**) Protein abundance estimates based on spectral counts and protein length as a percentage of the total number of spectra identified per mass spectrometry run (dNSAF, see Methods) were used to calculate relative enrichment between the three mass spectrometry datasets. The ratio of dNSAF values for different VFPs between HSV-1 infected MMs and mock-infected MMs (**A**) and between NEs and mock-infected MMs (**B**) are shown. Only those VFPs that had a HSV-1 infected MMs:mock-infected MMs ratio of >1.3 are presented. (**C**) Domain structure of VAPB. The protein has an MSP domain that when mutated at P56S causes a variant of amyotrophic lateral sclerosis (ALS8), followed by a coiled-coil domain that precedes the C-terminal transmembrane domain. (**D**) Lysates from HSV-1 infected or mock-infected HeLa cells were analyzed by Western blot for VAPB at different times post-infection. Quantification on repeats for 3 time-points by Li-COR shown on the right revealed no change in VAPB levels during the course of infection.

**Figure 3 cells-08-00120-f003:**
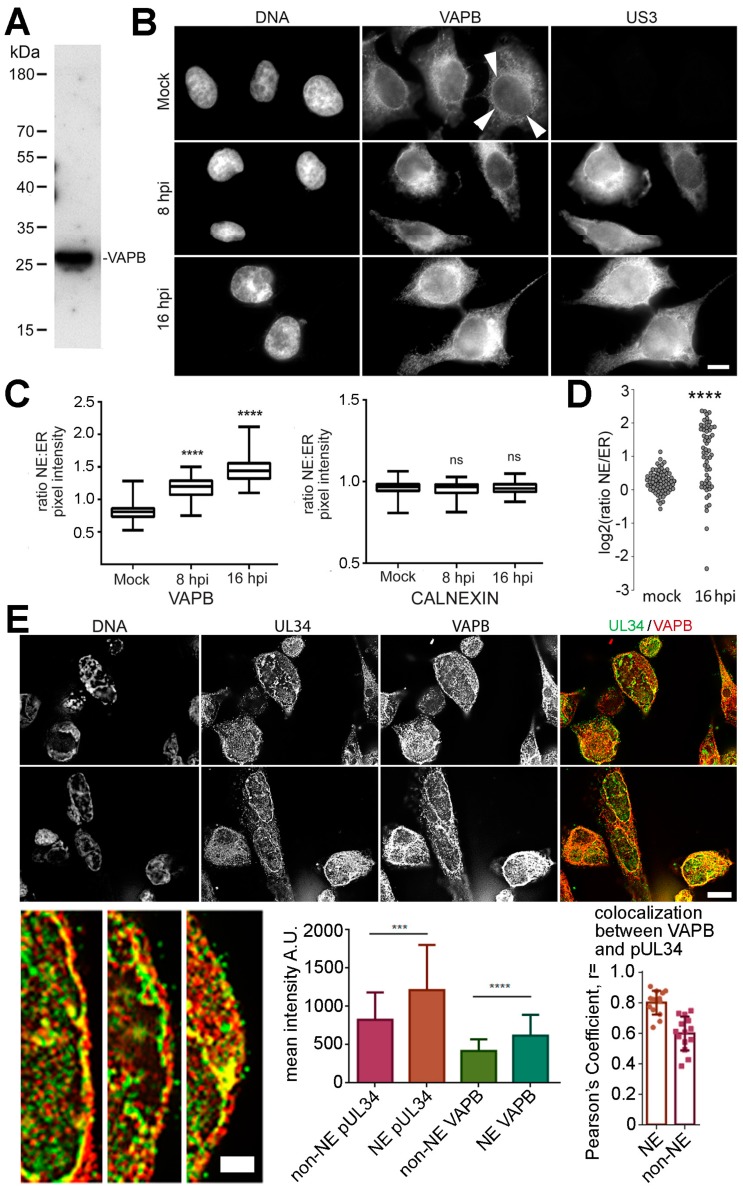
VAPB is at the NE during HSV-1 infection. (**A**) Confirmation of specificity of VAPB antibody by Western blot of HeLa cells. (**B**) HeLa cells were either mock infected or infected with MOI = 10 HSV-1 (WT). At 8 and 16 hpi, cells were fixed and processed for regular widefield immunofluorescence microscopy with VAPB mouse monoclonal antibodies and rabbit US3 antibodies (to identify infected cells). Some VAPB signal gave a distinct rim around the NE (indicated by arrowheads on one cell in the top VAPB panel), though signal also appeared throughout the ER. Scale bar, 10 µm. (**C**) The relative pixel intensities in the ER and NE were quantified. For each cell, five lines were drawn through the middle of the nucleus and pixel intensity was measured at a point in the nuclear rim (based on DAPI staining) and at a point 2 µm distant into the ER and the NE/ER ratio was calculated. Boxplots from 30 cells are shown in the graph below the images with the median (central line) and the error bars (grey) marked. Each sample was compared with its control (mock cells) by one-way ANOVA analysis followed by the Holm-Sidak’s multiple-comparison test. Significant *p*-values (**** *p* ≤ 0.0001) illustrate the general trend of these vesicle fusion proteins to accumulate at the NE upon infection. (**D**) Separately, after defining the NE in relation to the DAPI signal, the total NE fluorescence and all fluorescence signal outside the nucleus was quantified. From this data, mean fluorescence intensities from the whole NE and ER in sections were quantified, the ratios of NE:ER signal were determined, and their distribution was plotted using a log scale. This further revealed a wide distribution of NE:ER ratios in the infected cells compared to a tight distribution for the mock infected. The shift change in distribution with HSV-1 infection was still significant using a pair-wise Dunn test: **** *p* < 0.0001. (**E**) Microscopy images of cells co-stained with VAPB and pUL34 antibodies. Z-stacks of images were taken using 0.2 µm steps and then deconvolved. Images shown are from individual sections. Zoom images are shown in the bottom left corner of the panel with the scale bar for the upper image 10 µm and that for the zoomed images 2.5 µm. The first graph is from quantifying the mean pixel intensity in the NE compared to that in all other regions of the cell (including the nuclear interior), using the DAPI stained DNA to define the nuclear edge. The standard deviation of the mean is shown and paired *t* tests confirmed significance: *** *p* < 0.001; **** *p* < 0.0001 The graph in the right corner plots the Pearson’s Correlation Coefficient for the overlap between VAPB and pUL34 signal in the NE and in the other regions of the cell. Standard deviations are shown along with the distribution of values.

**Figure 4 cells-08-00120-f004:**
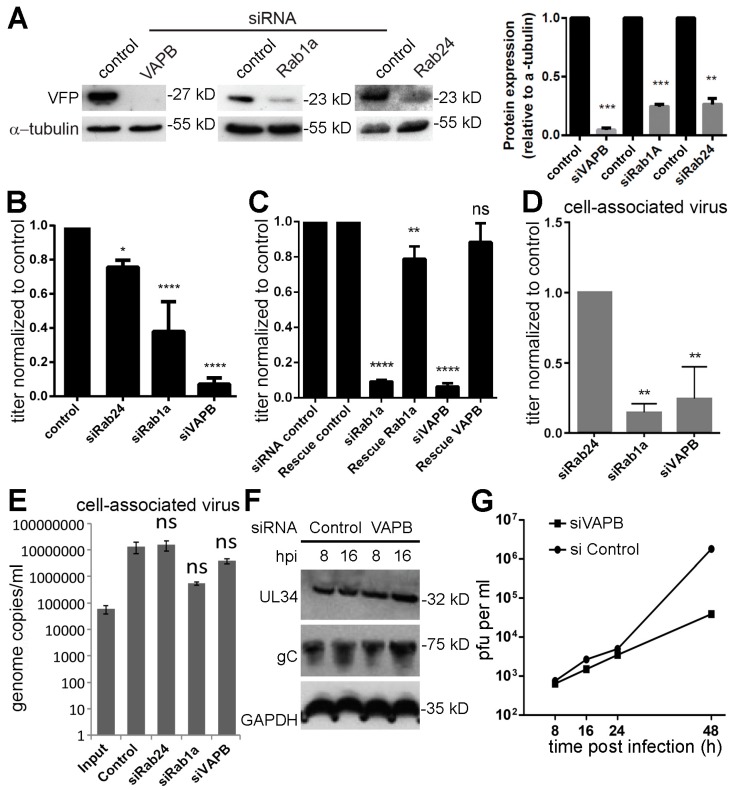
VAPB knockdown inhibits HSV-1 infection. (**A**) HeLa cells were transfected with siRNA oligos for VAPB and controls and cells lysed after 48 h. Western blot confirmed knockdowns. Quantification from three experiments is given in the right panel: error bars are to the standard deviation and the *p*-values using a one-way ANOVA and multiple comparisons by Dunnett’s test for comparing each condition with the control are shown (** *p* ≤ 0.01; *** *p* ≤ 0.001; **** *p* ≤ 0.0001). (**B**) HeLa cells were transfected with the control, Rab24, Rab1a, or VAPB siRNAs. After 48 h, cells were infected with HSV-1 (MOI = 10) and supernatant virions were collected at 16 hpi. Titers were established on U2OS cells and bars represent the average of three independent experiments, normalized to the control siRNA. Error bars are to the standard deviation and the *p*-values using a one-way ANOVA and multiple comparisons by Dunnett’s test for comparing each condition with the control are shown (* *p* ≤ 0.05; **** *p* ≤ 0.0001). (**C**) Rescue experiments with cells expressing wild-type protein resistant to the siRNAs were performed and supernatant virions also collected at 16 hpi. In all cases, the cells carrying both the rescue plasmid and the siRNAs recovered to roughly 80–90% of the control virus titers. Error bars are to the standard deviation and the *p*-values using a one-way ANOVA and multiple comparisons by Dunnett’s test for comparing each condition with the control are shown (** *p* ≤ 0.01; **** *p* ≤ 0.0001). (**D**) After 48 h, HeLa cells transfected with the siRNAs were infected with HSV-1 (MOI = 10) and cell pellets collected at 16 hpi. Titers were established on U2OS cells and bars represent the average of three independent experiments, normalized to the data for Rab24 siRNA, used as a control to better parallel background knockdown effects compared to non-target siRNA. Error bars are to the standard deviation from two separate experiments and the *p*-values using a pair-wise test (Tukey) to compare each condition with the Rab24 control are shown (** *p* ≤ 0.01; specifically for Rab1a *p* = 0.001938 and for VAPB *p* = 0.003162). (**E**) Total genome copies for cell-associated virus were determined by qPCR. (**F**) Levels of HSV-1 pUL34 and gC1 protein were determined by Western blot between the control siRNA and VAPB knockdown cells. The housekeeping protein GAPDH was used as a loading control. (**G**) HeLa cells were transfected with either a control siRNA or siRNA against VAPB. A multistep HSV growth curve over 48 h was performed by infecting cells at MOI = 0.1 in triplicate. Samples were harvested at the indicated times. Titers were determined by plaque assay using U2OS cells.

**Figure 5 cells-08-00120-f005:**
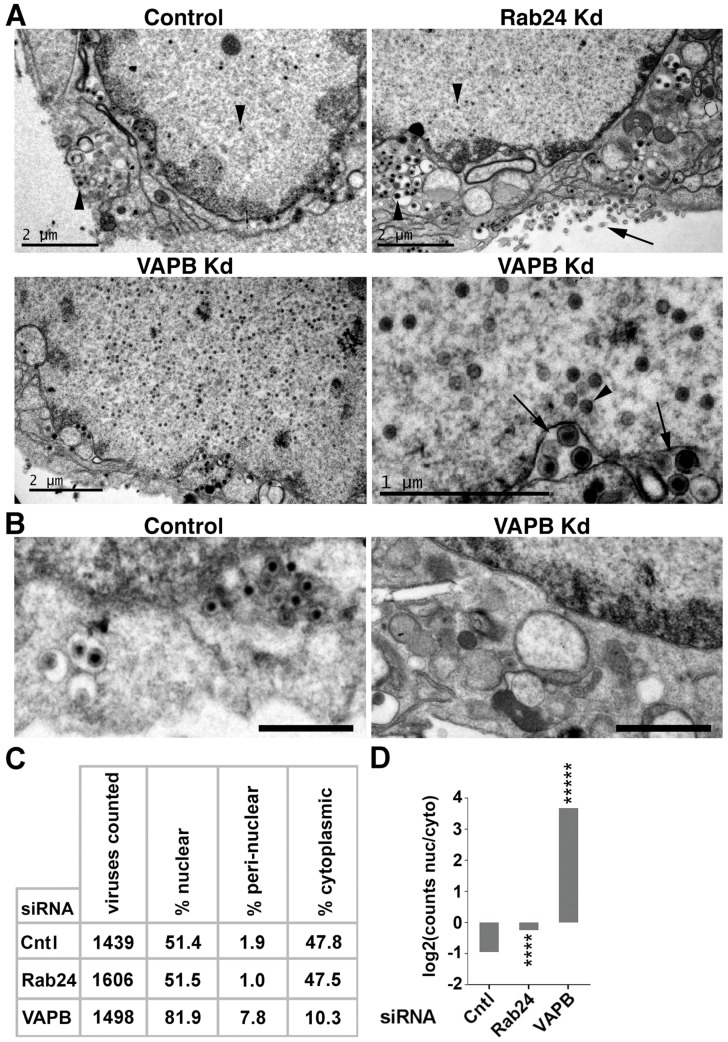
Electron microscopy reveals accumulation of virus particles in the nucleus and their relative absence from the cytoplasm with VAPB knockdown. In all images, the nucleus is at the top of the panel. (**A**) In control and Rab24 knockdown cells, some non-enveloped virus particles could be observed in the nucleoplasm (one example shown for each upper image with downward facing arrowheads), but many enveloped and non-enveloped particles could also be observed in the cytoplasm (one example shown for each upper image with upward facing arrowheads) as well as released mature particles just outside the cell (a large accumulation of these is highlighted with the angled arrow in the upper right image). In contrast, for the VAPB knockdown, few particles could be seen in the cytoplasm and visibly more nucleoplasmic non-enveloped particles were observed. In a zoomed image, several particles that had acquired a primary envelope were observed for the knockdowns in the NE lumen (two sets of multiple virus particles indicated by arrows). A single arrowhead highlights an assembled virus nucleocapsid prior to envelopment to contrast for particle size at the different stages. (**B**) Images focused more on the cytoplasm reveal virus particles in the cytoplasm for the siRNA control but a relative dearth of viral particles in the cytoplasm for the VAPB knockdown. Scale bars, 1 µm. (**C**) Over 1400 virus particles were counted for each condition and the percentages of nuclear, peri-nuclear (NE lumen), and cytoplasmic particles is given. Virus particles specifically accumulated in the nucleus and peri-nuclear lumen with VAPB knockdown. (**D**) Plotting the log2 ratio of nuclear:cytoplasmic virus particles for each condition revealed an inversion in the nuclear:cytoplasmic ratio where from particles being slightly more in the cytoplasm for the siRNA and Rab24 knockdown controls they became predominantly nuclear for the VAPB knockdown. Use of the Fisher’s exact test reveals very significant differences from the control (**** *p* < 0.0001; ***** *p* < 1 × 10^−50^).

**Figure 6 cells-08-00120-f006:**
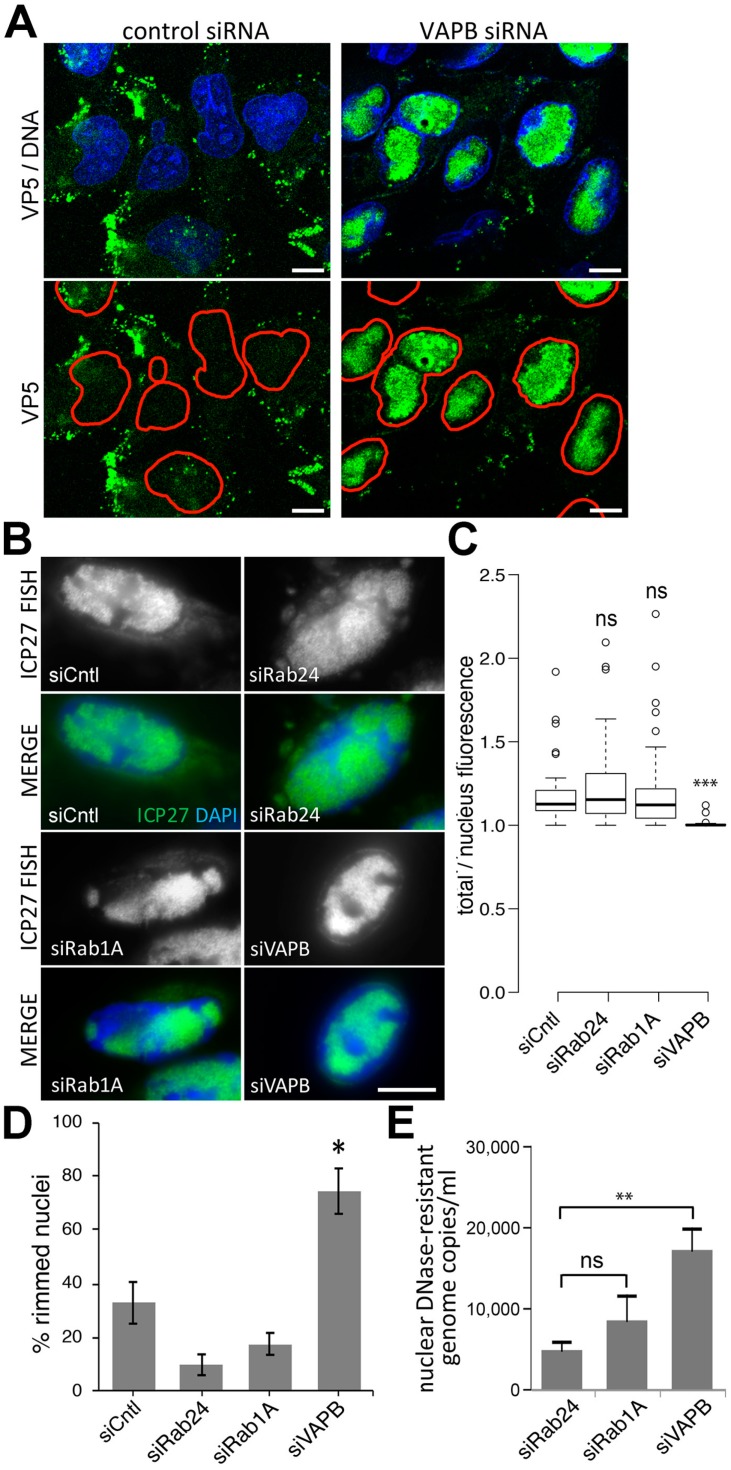
FISH and immunofluorescence also indicate accumulation of nuclear and peri-nuclear virus particles in VAPB knockdown cells. (**A**) Immunofluorescence staining for capsid protein VP5 at 16 hpi with an MOI = 10 reveals an altered distribution to acccumulate in the nucleus with VAPB knockdown. In the lower panels, the edge of the nucleus defined by DAPI staining of the DNA is marked by a red line so that a portion of the VP5 in the nucleus can be observed in the control knockdown cells, as expected. (**B**) The virus gene ICP27 was used as a FISH probe and labeled with biotin. Cells were knocked down for VAPB and controls Rab24, Rab1A and non-target siRNA, infected with HSV-1 and at 16 hpi fixed and processed for FISH. The hybridized virus ICP27 gene was visualized with streptavidin conjugated to Alexa488 dye and imaged by immunofluorescence microscopy. Cells were co-stained with DAPI to identify the nucleus. Images were taken at the midplane of the nucleus to minimize any signal from above or below the nucleus. This tends to also maximize the sectioned area of the ER/Golgi. Scale bar, 10 µm. (**C**) Using the DAPI nuclear staining to generate a mask of the nuclear area, the nuclear pools of hybridized virus ICP27 DNA were quantified from roughly 50 cells for each condition. Our previous studies on genome organization have shown that this 2D analysis is sufficient to see differences revealed by more intensive 3D imaging. The total hybridized ICP27 DNA in the same cell was also quantified and plotted divided by the nuclear signal so that values above 1 reflect the cytoplasmic pool of viral genomes. A clear increase in cytoplasmic viral genomes can be seen for the non-target siRNA control, the Rab24 and the Rab1A knockdowns, while no notable increase in cytoplasmic viral genomes was observed for VAPB knockdown. Statistical measurements were performed using a 2-tailed ANOVA analysis: *** *p* < 0.001. (**D**) The same images of cells analyzed for FISH were also counted for the visible accumulation of viral genome signals at the NE. The percentage is plotted and statistical significance from Fisher’s Exact test is given as: * *p* < 0.05. (**E**) In a separate experiment, DNase-resistant viral genomes were isolated by preparing nuclei from HSV-infected cells (MOI = 10, 16 hpi) following siRNA knock down. Nuclei were disrupted and incubated with DNase to remove unencapsidated DNA. DNA was prepared using a QIAamp DNA purification kit with RNase treatment to remove viral mRNAs. Remaining viral DNA was quantified by qPCR. Increases in encapsidated viral DNA of greater than 4-fold were observed upon siRNA depletion of VAPB, but not the control, Rab24. The data are from a single experiment and standard deviations and statistics are generated from three technical replicates with an ANOVA of *p* = 5.6 × 10^−05^: pair-wise Tukey test ** *p* < 0.01.

**Figure 7 cells-08-00120-f007:**
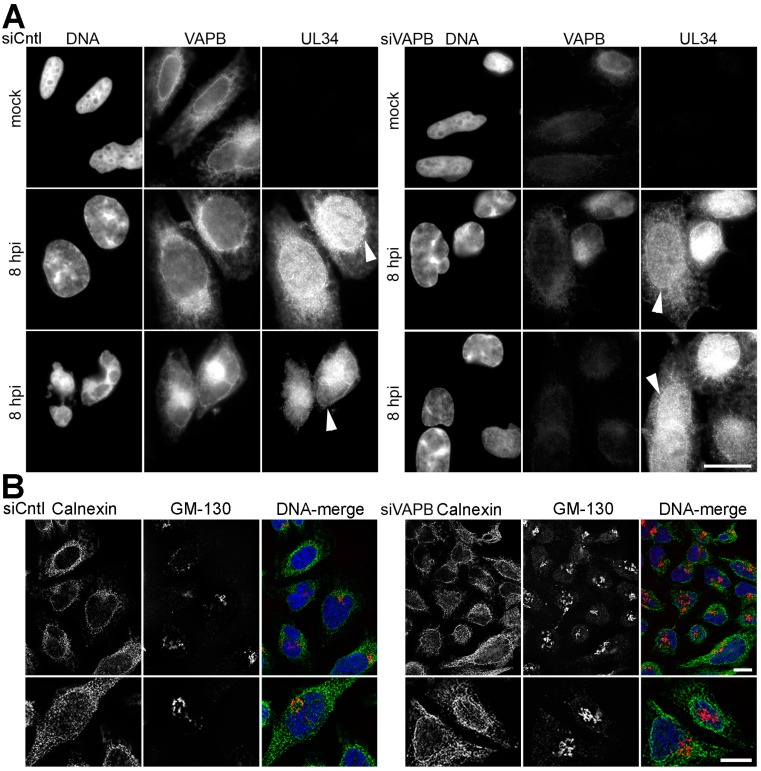
VAPB knockdown neither alters distribution of pUL34 nor notably disrupts other cellular membranes. (**A**) HeLa cells were knocked down for non-target control (Cntl) or VAPB. Cells were stained for confirmation of the knockdown and for pUL34. pUL34 staining could still be observed at the nuclear rim with the VAPB knockdown at 8 hpi. Two separate images are shown to highlight that an increased concentration of staining at the NE (defined as the edge of the DNA staining) is observed in both flatter cells with more normal looking nuclei and more rounded cells with distorted nuclei within the same population. Arrowheads are shown in some cells to indicate the NE. (**B**) Staining Cntl or VAPB knockdown cells for the ER marker calnexin and the Golgi marker GM-130 revealed no notable differences in staining with VAPB knockdown. Bottom panels are zoomed regions from other images from the same condition. Scale bars, 10 µm.

**Figure 8 cells-08-00120-f008:**
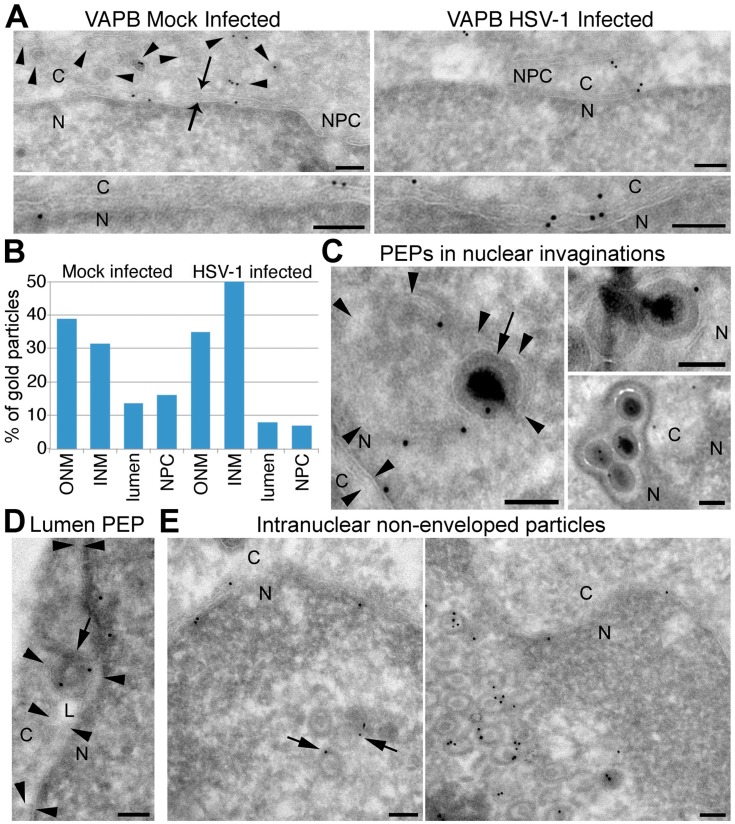
VAPB accumulates in both the ONM and INM and associates with virus particles. HeLa cells, either mock infected or infected with WT HSV-1 MOI = 10, were fixed at 16 hpi and cryosectioned prior to labelling with VAPB mouse monoclonal antibodies and anti-mouse conjugated gold particles for electron microscopy. (**A**) Mock infected cells (left panels) and HSV-infected cells in areas lacking virus particles (right panels) to investigate the membrane localizations of VAPB. In all panels, the cytoplasm is on the top and the nucleoplasm is on the bottom with this labeled in the upper left panel by C and N, respectively, and nuclear pore complexes in the top panels are labelled NPC. The upper left panel also contains two arrows that face towards the ONM and INM. A subset of visible ER membranes, many in association with gold particles, are indicated with arrowheads. (**B**) Quantification of 206 and 163 VAPB-labelled gold particles at the NE from respectively the mock infected and HSV-1 infected populations using 81 images each. The percentage of total NE gold particles associated with different parts of the nuclear membrane are plotted. Notably, a reasonable number of VAPB-labelled gold particles were observed in all membrane compartments: ONM, INM, NE lumen, and pore membrane with the NPC. Note that counted luminal particles may reflect sectioning angles where the gold particle artificially appears to be on the other side of the membrane, but as this cannot be ascertained for certain, they were counted separately. (**C**) Left: high magnification images of primary enveloped HSV-1 particles (PEPs) with VAPB-labelled gold particles that appear to be in membrane-bound invaginations of the NE. In the left image, the arrow points to the viral envelope and the arrowheads point to areas where the membrane of the invagination can be distinguished with N and C demarcating the nuclear and cytoplasmic compartments. The invaginated membrane also is associated with gold particles. (**D**) An enveloped HSV particle with VAPB-labelled gold particles clearly inside the NE lumen. In addition to the N and C demarcating the nuclear and cytoplasmic compartments, the INM and ONM are also highlighted by arrowheads facing one another. (**E**) Nuclear virus particles that appear to lack an envelope in association with VAPB-labelled particles. The sections have the appearance of being glancing sections at the surface of the nucleus. The two arrows in the left panel indicate gold particles associating with virus particles. All scale bars are 100 nm.

**Table 1 cells-08-00120-t001:** Abundance of viral proteins identified in HSV-1 infected MMs.

Protein	Unique Peptides	Spectral Counts	dNSAF Score	ID’d in Padula Study
US6 Glycoprotein D	7	38	0.00303	YES
UL27 Glycoprotein B	14	62	0.00216	NO
UL44 Glycoprotein C	6	30	0.00185	NO
Nuclear egress membrane protein pUL34	4	15	0.00171	YES
Virion protein US2	4	14	0.00151	NO
Membrane protein UL45	2	7	0.00128	NO
UL49 Tegument protein VP22	4	11	0.00115	YES
UL19/VP5 Major capsid protein	18	46	0.00105	YES
UL48 Tegument protein VP16	6	16	0.00103	NO
UL50 Deoxyuridine triphosphatase	3	12	0.00102	NO
US8 Glycoprotein E	8	16	0.00091	NO
UL18/VP23 Capsid triplex subunit 2	4	9	0.00089	YES
UL46 Tegument protein VP11/12	7	19	0.00083	NO
UL47 Tegument protein VP13/14	7	14	0.00063	NO
UL10 Glycoprotein M	1	8	0.00053	NO
UL42 DNA pol processivity subunit	4	8	0.00052	NO
US7 Glycoprotein I	3	6	0.00048	NO
UL40 Ribonucleotide reductase subunit 2	3	5	0.00046	NO
Tegument protein UL7	3	4	0.00042	NO
UL22 Glycoprotein H	5	11	0.00041	NO
pUL31 Nuclear egress lamina protein	2	4	0.00041	NO
UL29/ICP8 Single-stranded DNA binding protein	7	15	0.00039	NO
UL39 Ribonucleotide reductase subunit 1	8	14	0.00039	NO
Tegument protein UL51	2	3	0.00039	NO
ICP4	10	15	0.00036	NO
US10	2	3	0.00030	NO
UL12 Deoxyribonuclease	4	6	0.00030	NO
UL54 Multifunctional expression regulator	3	4	0.00025	NO
UL24	1	2	0.00023	NO
US1/ICP22	2	3	0.00023	NO
UL41 Tegument host shutoff protein	2	3	0.00019	NO
Tegument protein UL21	1	3	0.00018	NO
UL26/VP24/VP21 Capsid maturation protease	2	3	0.00015	YES
UL39/VP19C Capsid triplex subunit 1	1	2	0.00014	YES
Tegument protein UL25	2	2	0.00011	NO

**Table 2 cells-08-00120-t002:** Peptide and spectral counts for VAPB proteins passing the candidate criteria.

	HSV-1 Infected MMs				Mock Infected MMs				Mock Infected NEs		
Protein	Unique Peptides	Spectral Counts	dNSAF	HSV1 MMs:mockMMs Ratio	Unique Peptides	Spectral Counts	dNSAF	MockNEs:MockMMs Ratio	Unique Peptides	Spectral Counts	dNSAF
VAPB	2	7	0.002286	6.3	1	3	0.000363	9.0	2	8	0.003261
VAMP7	2	4	0.000702	4.9	1	2	0.000142	17.8	3	11	0.002523
RAB11B	7	49	0.007065	2.0	6	61	0.003560	3.4	7	65	0.012242
RAB9A	2	3	0.000469	1.9	1	4	0.000253	4.8	3	6	0.001226
RAB1A	4	35	0.005357	1.7	5	50	0.003093	11.9	5	184	0.036801
RAB18	8	23	0.003509	1.7	7	33	0.002038	22.1	8	226	0.045044
STX7	1	2	0.000241	1.7	1	3	0.000146	12.9	4	12	0.001888
RAB2A	4	17	0.002520	1.4	6	30	0.001800	6.1	9	57	0.011039
RAB35	3	17	0.002658	1.4	4	30	0.001899	1.4	3	13	0.002655

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
