# Peer review of "Host Vesicle Fusion Protein VAPB Contributes to the Nuclear Egress Stage of Herpes Simplex Virus Type-1 (HSV-1) Replication"

_cells, 2019, doi:10.3390/cells8020120_

Round 1

Reviewer 1 Report

Translocation of mature herpesvirus capsids from the nucleus to the cytoplasm is accomplished by budding at and fission from the inner nuclear membrane resulting in enveloped nucleocapsids (=primary virions) located in the perinuclear cleft.  Two herpesviral proteins, UL31 and UL34 forming the nuclear egress complex (NEC), are the major actors in this process. However, how subsequent fusion of the primary virion envelope with the outer nuclear or the ER membrane is coordinated to finally release the nucleocapsid into the cytosol is still  unknown. Although several viral proteins were supposed to take part, none of them was found crucial, which led to the assumption that a cellular pathway might be utilized.

Here the authors raised the hypothesis that host proteins facilitating nuclear egress should enrich in the nuclear membranes but also by diffusion in the endoplasmic reticulum, which is contiguous with the outer nuclear membrane. To test this, they isolated nuclear and microsomal membranes (MM) from mock infected as well as MM from herpes simplex virus 1 (HSV-1) infected HeLa cells. Mass spectrometry was applied to identify proteins, which are present in the nuclear envelope in mock-infected cells and enriched in MMs of HSV-1 infected cells versus non-infected cells.  Gene ontology analyses uncovered several proteins associated with vesicle-mediated transport. The role of one of the most enriched proteins, vesicle-associated protein B (VAPB), during HSV-1 infection was further investigated.

Although the approach is very interesting and new, the data presented here appear too preliminary and require additional experimentation. It was not previously suggested or supported by any experimental data that proteins involved in herpesvirus nuclear egress will end up in the ER. Even the fate of the viral nuclear egress proteins after nuclear egress has occurred is unclear. One very crucial Western blot experiment is therefore missing proving the presence of UL34 in infected MMs, which should accumulate together with cellular proteins in the MM according to the authors’ hypothesis. Moreover, herpesviruses are known to cause Golgi-fragmentation and disturbance of normal trafficking (e.g. Campadelli et al., 1993), which might at least to some degree account for the differences in protein abundance.  

Immunofluorescence images throughout the manuscript need higher resolution and magnification to better support the conclusion. The drop of viral titers after infection of VAPB knockdown cells might be due to defects throughout the herpesvirus replication cycle, which needs to be more carefully tested (virus entry, viral protein expression or trafficking, etc.). In addition, a very crucial control of the membrane fractionation experiments and MS, using a virus mutant defective for nuclear egress, e.g. lacking one of the NEC components, would increase the impact of the manuscript considerably.

Major comments:

Fig1 C: It is not completely clear whether these represent membrane preparations from infected or mock-infected cells. It seems that this is non-infected, but infected MM preparations should be included. Moreover, if the basic hypothesis is correct that proteins involved in nuclear egress turn up in the MM fraction, viral UL34 should show up. A western blot experiment could clarify this point. Cellular markers used in D as well as other viral components, e.g. glycoproteins should be also tested. This could should shed some light on the impact of infection on infected cellular membrane composition.

Fig2: It should be at least mentioned why VAMP7 was not further addressed – it is highly enriched in the nuclear envelope compared to MMs in mock infected cells (18-fold) and is enriched in the HSV-1 infected MMs (similar as VAPB) compared to mock MM. With this, VAMP7 would be even better fit into the authors hypothesis.

Fig3: Rim staining of VAPB or colocalization with UL34 is not evident in the images presented. In addition, ER staining should be added to better visualize the difference in pixel intensities measured in panel C. Infected cells usually become more spherical and might additionally influence the staining pattern.

Fig4 E: The difference in virus titers in the VAPB knockdown cells is evident only at very late time points after infection with a low moi. If knockdown of VAPB leads to a delay in nuclear egress, the effect should be detectable (predominantly) at early time points. How does this fit to the 2-log difference presented in panels B-D at 16 hrs p.i.?

Fig7 A: It is surprising that no VP5 signal is left in the nucleus in the control cells. Is protein expression comparable in infected control and VABP knockdown cells? B: Why was ICP27 “fish” done, despite having a virus expressing ICP27-GFP (as measure for immediate early protein expression) as well as VP26-RFP (RFP labelled capsids) (lines 76-77), which could be used as nucleocapsid marker in the cytoplasm? Viral DNA in the “fish” should be detectable in discrete foci (=nucleocapsids) the cytoplasm – the signal here in contrast appears diffuse and unspecific.

Fig9: Assigning gold particles to specific structures, especially for those closely together as the two nuclear membranes is error-prone. Although not described in any detail, immune labeling was probably done with a gold-labeled secondary antibody labelling the VAPB-specific antibody, which locates the gold particles at some distance from its antigen (length of two antibodies). In addition, to judge specificity of the label, images showing cytoplasmic membranes as positive controls should be added. Moreover, many gold particles were found associated with immature B-type capsids (panel E). How does this fit into the hypothesis? 

Minor remarks:

Line 24: it is not clear whether the authors tested indeed for differences in cell-associated versus released virus particles; this should be added to the Mat/Met section;

Line 54: it should read ICP34.5

Line 78: Was the virus indeed adsorbed to the cells at 37oC? This should already allow for penetration into cells.

Line 77: The ICP27-GFP expressing virus seems not to be used in the study;

Line 80: the starting number of cells for the membrane preparation should be given;

Fig.1 A: Western blot experiments showing expression of UL31 and (or) UL34 would be very helpful;

Fig1 B: Are these images taken 16 or 10 hrs p.i. as mentioned in the text?

Line 156- 160: Is it indeed surprising that proteins with “vesicle-mediated transport” functions are found enriched in the MM fraction?

Fig2C: The control Western blot using anti-tubulin does not fit to the VAPB specific blot – the correct control should be shown; at best on the same blot

Fig.4E: cells for panel E were infected with a Moi of 1 or 0.1 as stated in the figure legend?

Fig6: Counting indicates a nearly 50:50 distribution of nuclear and cytoplasmic capsids in control cells and not as stated a “predominant cytoplasmic”.

Fig7 A: Which time point and MOI was used?

Fig8: Nuclear rim staining of UL34 is not detectable in the images.

Line 353: IF images shown in the manuscript seem not derived from a confocal plane. The microscope mentioned in Mat/Met at least seems not to provide the technique;

Author Response

Responses in blue beneath each reviewer point

Comments and Suggestions for Authors

Translocation of mature herpesvirus capsids from the nucleus to the cytoplasm is accomplished by budding at and fission from the inner nuclear membrane resulting in enveloped nucleocapsids (=primary virions) located in the perinuclear cleft.  Two herpesviral proteins, UL31 and UL34 forming the nuclear egress complex (NEC), are the major actors in this process. However, how subsequent fusion of the primary virion envelope with the outer nuclear or the ER membrane is coordinated to finally release the nucleocapsid into the cytosol is still  unknown. Although several viral proteins were supposed to take part, none of them was found crucial, which led to the assumption that a cellular pathway might be utilized.

Here the authors raised the hypothesis that host proteins facilitating nuclear egress should enrich in the nuclear membranes but also by diffusion in the endoplasmic reticulum, which is contiguous with the outer nuclear membrane. To test this, they isolated nuclear and microsomal membranes (MM) from mock infected as well as MM from herpes simplex virus 1 (HSV-1) infected HeLa cells. Mass spectrometry was applied to identify proteins, which are present in the nuclear envelope in mock-infected cells and enriched in MMs of HSV-1 infected cells versus non-infected cells.  Gene ontology analyses uncovered several proteins associated with vesicle-mediated transport. The role of one of the most enriched proteins, vesicle-associated protein B (VAPB), during HSV-1 infection was further investigated. 

Although the approach is very interesting and new, the data presented here appear too preliminary and require additional experimentation. It was not previously suggested or supported by any experimental data that proteins involved in herpesvirus nuclear egress will end up in the ER. Even the fate of the viral nuclear egress proteins after nuclear egress has occurred is unclear.

We are surprised by this comment and not certain what additional experimentation the reviewer is asking for beyond the knockdown rescue experiments with virus titres and genomes quantified multiple ways, electron microscopy quantification of virus particles in the nucleus, nuclear envelope lumen and cytoplasm, immunofluorescence and immunogold electron microscopy experiments we have already done.  The standard publishing approach for screens is to identify a number of candidates from the screen and then characterize just one of them in detail, which is what we have done here. If the reviewer is wanting additional support for the premise of the screen then the principle of transmembrane protein translocation between the ER and the outer and inner nuclear membranes goes back really to the 1990 Powell and Burke heterokaryon experiment that was quickly followed by a more direct test of translocation in the Soullam and Worman 1993 and 1995 J Cell Biol papers.  Since then, this area has been extensively investigated using more creative modern approaches and technologies by the Gerace (Ohba et al., 2004 J Cell Biol), Blobel (King et al., 2006 Nature), Schirmer (Zuleger et al., 2011 J Cell Biol), Veenhoff (Meinema et al., 2011 Science), Ellenberg (Boni et al., 2015 J Cell Biol), Kutay (Ungricht et al., 2015 J Cell BIol) labs as well as several others. Thus, the premise for the screen is based on a wealth of previous data and solid principles. However, we recognize that the virus community is likely not acquainted with this literature and so have changed the text in the introduction from:

"During HSV infection, any host proteins facilitating egress might be expected to be found in both the NE and, due to its contiguity with the NE, also in the ER where they might diffuse after fusion with the ONM."

to:

"It has clearly been established that transmembrane proteins can translocate between the nuclear membrane and the ER by a lateral diffusion mechanism (25-28). Due to there being both many connections between the outer nuclear membrane (ONM) and the ER and between the ONM and the nuclear pores, a protein in the ONM could diffuse either to the ER or the inner nuclear membrane. Therefore, during HSV infection, any host proteins facilitating egress that fuse with the ONM might be expected to be found in both the NE and the ER where they moreover might be observed to increase in abundance."

and in the beginning of the results from:

"Cellular proteins involved in nuclear egress should accumulate in the NE and possibly also in the contiguous ER after fusion with the ONM."

to

"Cellular proteins involved in nuclear egress must be present and might even accumulate in the NE during infection. Moreover, if they become part of the primary envelope and/or are involved in de-envelopment fusion with the ONM, they may specifically accumulate in the ONM as nuclear egress progresses and possibly also in the contiguous ER after fusion with the ONM."

to better explain this point.

Nonetheless, there is no experiment that can unequivocally prove that a specific protein is moving from the NE to the ER as opposed to having been generated by de novo synthesis in the ER other than generating a photoactivatable-GFP fusion and activating it in the nuclear envelope to then follow it moving to the ER. Unfortunately, this experiment is not possible for pUL34 because many labs have previously shown that tagging this protein alters its function.  Nonetheless, due to the virion host shut-off function of HSV which acts early in infection, it is unlikely that at this timepoint in infection all the host nuclear envelope proteins that increased abundance in the infected microsomes would be due to de novo synthesis.

With regards to the statement that there is no data here showing that host or viral nuclear egress proteins accumulate in the ER during infection, there certainly was none and that is part of what we are testing here and demonstrated in the data in the Supplementary Table that has now been moved to a new Table 1 (see reply to next comment).  Thus we have changed the sentence towards the end of the introduction from "To search for candidate viral fusion facilitators, these membrane components were prepared ..." to " To test this and to search for candidate viral fusion facilitators, these membrane components were prepared ...". Nonetheless, it is not possible, as noted above for pUL34, to distinguish protein that participated in nuclear egress to de novo synthesized protein and so we have explained and qualified this in the text and we would argue that with all the novel data already in this paper that to be asked to also show something that may not be possible to test with current technologies even if it normally occurs is inappropriate.

One very crucial Western blot experiment is therefore missing proving the presence of UL34 in infected MMs, which should accumulate together with cellular proteins in the MM according to the authors’ hypothesis.

We do not understand why this is being requested as the mass spec data we show in Supplementary Table 1 clearly shows all the proteins previously identified in the primary envelope (Padula et al., 2009 J Virol) in the microsome (ER) fraction of the infected cells and pUL34 is reasonably abundant amongst them.  Mass spec data with multiple peptides is much more reliable than Western blots where antibodies often exhibit some cross-reactivity.  Nonetheless, we tried to additionally perform a Western blot for pUL34 as we were able to find some tubes remaining of the original samples.  Unfortunately, however, after running out all the tubes we found in the freezer it was clear that the samples had degraded since the experiment was done several years ago.  To repeat this experiment again when the mass spec data was already in the paper would not be practicable as recovery is very limited for microsome preps from tissue culture cells and so we had to grow 15 roller bottles of infected cells for the original experiment. 

As we fear that some readers might not check out the supplemental table we have decided to add this data into the paper as Table 1 in the text and change the existing Table 1 to Table 2.

Moreover, herpesviruses are known to cause Golgi-fragmentation and disturbance of normal trafficking (e.g. Campadelli et al., 1993), which might at least to some degree account for the differences in protein abundance.  

We are indeed aware of the early Roizman and Campanelli study (Avitabile et al., 1995 J Virol) showing Golgi structural alterations as early as 4 hpi as well as more recent studies such as the Martin et al., 2017 Front Cell Infect Microbiol study showing the dependence of these changes on Src kinase.  However, we stated in the text that increased abundance in the microsomes in of itself does not prove an involvement in nuclear egress as there are other possible reasons for changes in ER protein abundance during HSV-1 infection and this is only one of many.  The mass spectrometry screen was to identify a candidate pool from which proteins could then be tested.  The accumulation of virus particles in the nucleoplasm and their exclusion from the cytoplasm with VAPB knockdown shows that the defect here at least does not directly involve the Golgi.  We showed moreover in Figure 8B that there is no visible disruption to the Golgi from the VAPB knockdown from images using Golgi marker GM-130 +/- VAPB knockdown.  In addition, we observed very low abundance of virions in the cytoplasm of infected cells with VAPB knockdown indicates that HSV-1 effects on cytoplasmic trafficking may not be relevant for nucleus-confined virions.  Therefore we have not changed the text with regards to this point other than to add to the discussion the lines: "however, it is also possible that HSV-1 disruption of ER and Golgi membranes unintentionally results in a redistribution of VAPB where it can more indirectly contribute to nuclear egress and it remains possible that the trafficking of some important player in nuclear egress besides pUL34 is not able to gain access to the NE in the VAPB knockdown cells."

Immunofluorescence images throughout the manuscript need higher resolution and magnification to better support the conclusion.

We have now shown zoomed images in some figures and added more quantifications to Figure 3.

The drop of viral titers after infection of VAPB knockdown cells might be due to defects throughout the herpesvirus replication cycle, which needs to be more carefully tested (virus entry, viral protein expression or trafficking, etc.).

First, regarding possible defects in the replication cycle, we have tested total genome copies in cells and the knockdown cells are clearly replicating genomes similar to the control with only siRab1a having a visible reduction that was nonetheless not statistically significant. Thus, the reduction in virus titers appears to be due to the failure of the viral genomes to get out of the nucleus as opposed to a reduction in generation of viral genomes.

Second, regarding viral protein expression, we tested for levels of pUL34 and gC by Western in infected control and VAPB knockdown cells.  There is no visible reduction in the levels of these two proteins.  This data has now been added as panel F in Figure 4.  Though the data is from separate experiments and associated controls that would not make for a figure, we also have data on ICP27, gD and VP26 that similarly indicates their levels are not dropping in the knockdown cells.

Third, regarding effects on trafficking, we did not observe defects in ER staining with DiOC6 or calnexin staining or in Golgi staining with GM-130 staining in VPAB knockdown cells (the calnexin and Golgi staining are shown in Figure 7B, formerly 8B). We did not do trafficking assays as neither Schirmer nor Graham labs have expertise in this area. We acknowledge, however, that HSV-1 disruption of the Golgi and to a lesser degree ER membranes could influence the changes we observe for VAPB distribution during infection as opposed to an intentional recruitment mechanism and we have added a caveat about this into the discussion: "If VAPB interacts with virus proteins it could also provide a mechanism for specific recruitment of VAPB to egress complexes; however, it is also possible that HSV-1 disruption of ER and Golgi membranes unintentionally results in a redistribution of VAPB where it can more indirectly contribute to nuclear egress." Nonetheless, whether the virus specifically recruits VAPB, the knockdown of VAPB results in a defect in getting virus particles out of the nucleus and thus nuclear egress.

In addition, a very crucial control of the membrane fractionation experiments and MS, using a virus mutant defective for nuclear egress, e.g. lacking one of the NEC components, would increase the impact of the manuscript considerably.

We agree that this would have potentially been a nice extra sample to analyze and in fact had discussed this, but were unable to obtain such a mutant. Moreover, once we had begun these experiments and realized the massive scope of the infections needed to obtain enough MMs, we decided to limit the number of samples to test.  However, the reality is that while a virus mutant defective for nuclear egress might block the translocation and subsequent accumulation of cellular proteins in the ER, there are many instances where a different result would be expected due to the aforementioned lateral diffusion of proteins between the INM and ONM/ER through the peripheral channels of the NPC.  For example, if the virus protein important for nuclear egress binds to host proteins to corral them for supporting egress then the absence of the virus protein could also release the host protein so not having an INM tether it increases in the ER.  Considering the extreme amount of work involved in the experiment and its not being a conclusive control, we deemed it of too low priority to engage.

Major comments:

Fig1 C: It is not completely clear whether these represent membrane preparations from infected or mock-infected cells. It seems that this is non-infected, but infected MM preparations should be included.

Thank you for pointing out both the lack of clarity in the description and the absence of the third sample.  In the previous version of Figure 1B the MMs shown are infected.  In retrospect we don't know why the student did not include the mock MMs in the original figure, but we searched through her notebooks and gels and found another gel that had all three samples.  Thus in the new version of the figure we now show all three samples: mock-infected NE, mock-infected MMs, and HSV-1 infected MMs.

Moreover, if the basic hypothesis is correct that proteins involved in nuclear egress turn up in the MM fraction, viral UL34 should show up. A western blot experiment could clarify this point. Cellular markers used in D as well as other viral components, e.g. glycoproteins should be also tested. This could should shed some light on the impact of infection on infected cellular membrane composition.

As noted above, the mass spec data clearly show not only pUL34, but all the proteins previously identified in primary envelopes (Supplemental Table S1, and Padula et al., 2009 J Virol). To emphasize this point we have also added a new Table 1 into the text with this data.

We nonetheless considered running some Western blots as suggested to make the point more clearly also in the primary figure; however, though we found what was left of these samples in -20°C freezer, unfortunately the samples had degraded.  The time frame of revisions would not allow for another MM prep to be made and it does not seem so significant a point to justify generating another prep because due to the high losses of membrane in purifying the MMs we had to use 15 roller bottles of infected culture. 

Fig2: It should be at least mentioned why VAMP7 was not further addressed – it is highly enriched in the nuclear envelope compared to MMs in mock infected cells (18-fold) and is enriched in the HSV-1 infected MMs (similar as VAPB) compared to mock MM. With this, VAMP7 would be even better fit into the authors hypothesis.

As noted above the standard with papers containing screens is to choose one protein to follow up on.  We agree that VAMP7 is a particularly interesting candidate, especially as VAPB is known to interact with VAMP1 and 2 and so it might make sense for the two to interact in a complex; however, we chose VAPB initially for several reasons too numerous to mention. However, one aspect of VAPB makes it stand strongly above the rest in that it was recently reported to interact with the inner nuclear membrane protein emerin in the IntAct Molecular Interaction Database. Therefore, to give a better logic we have added to the text here "...and recently indicated to interact with the inner nuclear membrane protein emerin in the IntAct Molecular Interaction Database "https://www.ebi.ac.uk/intact/"."

At the same time, we agree with the reviewer that further mention of VAMP7 as a candidate is important to highlight. Therefore, we have added a paragraph to the discussion focused on the molecular interactions known for some of these VFPs that highlights the possibility that they could function in a complex together and here added the interaction of VAMPs with VAPB. The new paragraph states: "The demonstration of significant defects in virus nuclear egress for the VAPB knockdown suggests that other of the VFPs similarly highlighted by our study might also be involved in HSV-1 nuclear egress. This idea is more compelling by the general functioning of VFPs in larger complexes and observations that VAPB itself interacts with other proteins identified or closely related proteins. For example, Rab11, that was also highlighted by our mass spectrometery analysis, functions in a complex with VAPB and several other proteins (69, 70). Separately VAPB has been shown to interact with VAMP1 and 2 (51), both members of the same family as VAMP7 that was similarly highlighted in our mass spectrometry analysis. Moreover, the IntAct Molecular Interaction Database (71) reveals that VAMP7 is involved in several different vesicle membrane trafficking complexes including multiple SNARE complexes, one of which includes Stx7 that was also highlighed by our mass spectrometry analysis. Compellingly, IntAct also reveals VAPB interactions with the inner nuclear membrane protein emerin and multiple virus proteins from Hepatitis C Virus (72)."

Fig3: Rim staining of VAPB or colocalization with UL34 is not evident in the images presented. In addition, ER staining should be added to better visualize the difference in pixel intensities measured in panel C. Infected cells usually become more spherical and might additionally influence the staining pattern. 

We have occasionally found cells that give beautiful rim staining such as in the figure appended below; however, we feel that it is more accurate to show the more typical staining we get, hence the images we show in the manuscript. The fact is that these proteins are in both the NE and other cellular locations and we think it is misleading to just show the nicest images. We have quantified both NE and non-NE mean fluorescence intensity and Pearson's Correlation Coefficients for colocalization between VAPB and pUL34 in these different environments from the images shown in the paper and other similar images and now added this to Figure 3E. This unbiased data clearly shows increased mean pixel intensity at the NE and that there is more co-localization at the NE than in other parts of the cell. Regarding spherical cells, we have intentionally shown both flat and spherical cells (the showing of multiple images seems to have confused some reviewers) specifically so that the reader can see that there is some rim staining in both. We have tried to make this more clear in the text with this revision.

Fig4 E: The difference in virus titers in the VAPB knockdown cells is evident only at very late time points after infection with a low moi. If knockdown of VAPB leads to a delay in nuclear egress, the effect should be detectable (predominantly) at early time points. How does this fit to the 2-log difference presented in panels B-D at 16 hrs p.i.?

This is a multistep growth curve starting at MOI=0.1. So over the first hours only some cells get infected and there would be some virus production early on but as time progresses less and less virus is available to infect new cells. That is why there is a greater effect at late times.

Fig7 A: It is surprising that no VP5 signal is left in the nucleus in the control cells.

We have now modified Figure 7A to duplicate the panels and show underneath the VP5 signal without the DAPI staining of the DNA masking it. To mark the nuclear boundary a line was drawn demarcating the peripheral limit of the DAPI signal. This now clearly shows that there is also VP5 signal inside the nucleus for the control siRNA cells.

Is protein expression comparable in infected control and VABP knockdown cells? This refers back to the question above that WBs could sort out.

We have checked the expression of pUL34 and gC1 by Western blot in infected control and VAPB knockdown cells.  There is no visible reduction in the levels of these two proteins.  This data has now been added as panel F in Figure 4.

B: Why was ICP27 “fish” done, despite having a virus expressing ICP27-GFP (as measure for immediate early protein expression) as well as VP26-RFP (RFP labelled capsids) (lines 76-77), which could be used as nucleocapsid marker in the cytoplasm? Viral DNA in the “fish” should be detectable in discrete foci (=nucleocapsids) the cytoplasm – the signal here in contrast appears diffuse and unspecific. 

FISH was done as a quantitative way to measure total viral genomes, which neither of the labeled viruses mentioned would do. In general when using proteins as markers it is not always clear where it is accumulating free at ribosomes upon synthesis or assembled into capsids. Specifically for the VP26-RFP, note that the images in Figure 1B show a combination of diffuse and small punctate stainings in the nucleus and cytoplasm or very large accumulations that saturate the camera and make quantification between these states impossible.  Moreover, with herpesviruses there are a large number of defective viruses observed where genomes have not been encapsidated.  Finally, the criticisms levelled by reviewer 4 (who was under the misperception that we had used this virus for more than just Figure 1B as a way to follow the virus during infection while optimizing conditions) about ways that the RFP fusion can interfere with function render its use for this purpose a sub-optimal experiment. The ICP27 virus would similarly be a poor measure for this for slightly different reasons, but viewing the image we have provided roughly a page down in response to the query about line 77 should make it clear that proper quantification of the large accumulations using this virus would also not be possible.  

The use of our FISH approach where the signal is more diffuse and not saturated in few large accumulations is more quantitative, but it has its own limitations in that it does not discriminate between packaged genomes and recently replicated genomes that are not packaged.  However, note that we did this in addition to measuring by qPCR the nuclear DNase-resistant genome copies/ml in Figure 6E, which gives separate and additional information about how many genomes have been encapsidated. 

Regarding the statement that the FISH should look different, we assume this assumption is based on using fluorescent protein fusions to capsid proteins that tend to accumulate into aggregates and/or FISH images in Everett et al 2007 JVI 81:10991. Note that the author of that paper, our colleague Roger Everett was a very helpful advisor for our studies. Roger’s data were from cells very early in infection and he used Actinomycin D and cycloheximide to inhibit transcription and translation. This makes it easy to visualise punctate incoming viral genomes by FISH. We carried out FISH on untreated cells infected with HSV for 16 hours where the nucleus will be filled with non-specifically located viral genomes. Our FISH data were backed up by analysis of HSV encapsidated DNA and now also by VP5 staining to locate viral capsids in Figure 6A. Each approach gives a consistent result: retention of HSV-1 in the nucleus upon VAPB knock down.

Fig9: Assigning gold particles to specific structures, especially for those closely together as the two nuclear membranes is error-prone. Although not described in any detail, immune labeling was probably done with a gold-labeled secondary antibody labelling the VAPB-specific antibody, which locates the gold particles at some distance from its antigen (length of two antibodies). In addition, to judge specificity of the label, images showing cytoplasmic membranes as positive controls should be added. Moreover, many gold particles were found associated with immature B-type capsids (panel E). How does this fit into the hypothesis?  

Re the request for cytoplasmic membranes, these are indeed shown in panel A and the association of gold particles with ER membranes as expected from previous reports on VAPB is highlighted by arrows in the upper left panel.

Re the question about gold particles associated with immature B-type capsids, this actually fits well with the hypothesis as these images appear to show a clustering of virus particles under the membrane i.e. they could interact as part of a docking step supporting initial steps of egress to get into the NE lumen.

Minor remarks:

Line 24: it is not clear whether the authors tested indeed for differences in cell-associated versus released virus particles; this should be added to the Mat/Met section;

We have indeed tested both, but can see how the term "cell-released" might not be entirely clear. We had thought that while writing "supernatant" might be more clear to a virologist that as the readership of Cells will be more wide the term "cell-released" might be better. The reviewer's comment has caused us to reflect on this and we have decided to change this in the abstract to: "VAPB knockdown significantly reduced both cell-associated and supernatant virus titers." and in the Methods section 2.6 we added the sentence: "As indicated in the figure legends, either released virus was collected by pelleting from the supernatant or cell-associated virus was collected after extensive washing followed by pelleting cells." and finally in the figure legend we have also clarified that the rescue experiment in panel C was using supernatants by adding "... and supernatant virions also collected at 16 hpi."

Line 54: it should read ICP34.5

This has been corrected.

Line 78: Was the virus indeed adsorbed to the cells at 37oC? This should already allow for penetration into cells. 

Yes, cells were placed in the incubator and taken out frequently for shaking. This is standard textbook protocol for HSV infection (see Harland and Brown “HSV growth, preparation and assay. Chpt 1 in Herpes Simplex Virus Protocols 1998 eds. Brown SM & Mclean AR. Humana Press Totowa, New Jersey).  Recent data suggest that the virus gets in almost immediately, but incubating for an hour is known to allow optimum viral adsorption and we get closer to 100% infected cells in the culture.

Line 77: The ICP27-GFP expressing virus seems not to be used in the study; 

Our apologies.  We were initially using the ICP27-GFP virus to identify infected cells while eliminating the possibility of any antibody cross-reactivity in double labellings — despite that we were already using donkey minimal cross-reactivity secondary antibodies.  However, there was nothing particularly to learn from using this virus except as a staining control (see image below), which would be distracting to the reader, and so we did not use the images and forgot to take the virus out of the Methods.  We have now removed this.

Line 80: the starting number of cells for the membrane preparation should be given;

The prep was using 15 roller bottle cultures. We did not count cells throughout the protocol however, it is well known that this will provide, at 70% confluence, ~1.8 x 109 cells.  This information has been added to the Methods section. 

Fig.1 A: Western blot experiments showing expression of UL31 and (or) UL34 would be very helpful; 

As noted above, the lysates were degraded after several years in the -20°C freezer. The coomassie stained gel below shows that the MM samples were degraded so that we could not do this experiment without doing another 15 roller bottle prep, which is very costly and time-consuming. Thus, it did not seem worth doing due to the poor cost-benefit ratio when it would simply recapitulate data already in Table 1.

Again, the mass spec data clearly show both proteins in the infected MMs and this data has been highlighted in the form of a new Table 1.

Fig1 B: Are these images taken 16 or 10 hrs p.i. as mentioned in the text?

Thank you for pointing this discrepancy out.  The figure legend is correct in this case and we have now corrected the labelling in the Figure itself.

Line 156- 160: Is it indeed surprising that proteins with “vesicle-mediated transport” functions are found enriched in the MM fraction? 

Our statement did not use the word “surprising”. The statement was not about VFPs being enriched in MMs, but about their increasing in HSV-1 infected MMs over mock infected MMs. We simply pointed out that under high selection stringency VFPs made up to 20% of total proteins in the HSV-1-infected MMs.

We showed VFPs in the nuclear envelope fraction a number of years ago (Schirmer et al., 2003 Science) and Karen Oegema found Rab5 involved in nuclear envelope disassembly in C.elegans around the same time; so, yes, we fully agree that finding proteins with “vesicle-mediated transport” functions enriched in the MM fraction is not surprising. It is noteworthy, however, that even for many years after these papers the prevailing view in the community would have found this surprising. This has been changing in recent years, mostly due to several high profile reports of ESCRT proteins in the nuclear membrane published in recent years.

Fig2C: The control Western blot using anti-tubulin does not fit to the VAPB specific blot – the correct control should be shown; at best on the same blot 

We thank the reviewer for catching this oversight.  Since the experiment was repeated three times for the quantification we expect the student must have accidentally taken panels from two different repeats for the original figure.  We went through the student's notebook and Li-COR images and identified an experiment with both VAPB and tubulin channels labelled. We have now replaced the images with a matched set.

Fig.4E: cells for panel E were infected with a Moi of 1 or 0.1 as stated in the figure legend?

We apologise for the confusion and thank the reviewer for catching this. The correct number is MOI=0.1. This has now been changed in the text.

Fig6: Counting indicates a nearly 50:50 distribution of nuclear and cytoplasmic capsids in control cells and not as stated a “predominant cytoplasmic”.

We have changed the phrasing in the text from "...the expected distribution with some nuclear particles and many virus particles present in the cytoplasm at 16 hpi (Figure 5A)" to "...the expected distribution with many virus particles present both in the nucleus and in the cytoplasm at 16 hpi (Figure 5A)".  In the figure legend we have changed this from "(D) Plotting the log2 ratio of nuclear:cytoplasmic virus particles for each condition revealed an inversion in the nuclear:cytoplasmic ratio from particles predominantly in the cytoplasm for the siRNA and Rab24 knockdown controls and predominantly in the nucleus for the VAPB knockdown." to "(D) Plotting the log2 ratio of nuclear:cytoplasmic virus particles for each condition revealed an inversion in the nuclear:cytoplasmic ratio where from particles being slightly more in the cytoplasm for the siRNA and Rab24 knockdown controls they became predominantly nuclear for the VAPB knockdown."

Fig7 A: Which time point and MOI was used?

16 hpi, MOI=10. This is now stated in the figure legend.

Fig8: Nuclear rim staining of UL34 is not detectable in the images.

I believe the reviewer may be thinking that the entire pool of a protein must be in the NE to say that it has NE staining. This idea is rather outdated. In fact, there are a great number of studies recently demonstrating the concept of protein "moonlighting" e.g. that a protein can have multiple distinct cellular pools. In fact, several large-scale proteomic studies from Kathryn Lilley, Matthius Mann, and other labs have indicated that greater than 40% of all cellular proteins "moonlight". We have clearly written in the text that the protein is not only in the NE and even used terms like "weak rim staining". We have nonetheless added arrows now in both Figure 3 and Figure 7 (formerly Figure 8) to highlight the NE and we have performed quantification of signal intensities throughout the cells and added this to Figure 3 that clearly shows NE targeting.

Line 353: IF images shown in the manuscript seem not derived from a confocal plane. The microscope mentioned in Mat/Met at least seems not to provide the technique; 

Nearly all images shown in the manuscript were taken with the microscope described in the Methods; however, we had previously just indicated in the figure legends when deconvolution was additionally used.  We can see how this may have been confusing as we did not describe also the deconvolution setup in the Methods.  Also for Figure 6A (previously 7A), we had added this after a previous submission and failed to add to the Methods that it was taken using a different microscope in Glasgow. Thus we have changed the text in the Methods from "Images from the midplane of the nucleus were acquired on a Nikon TE-2000 microscope using a 1.45 NA 100x objective, Sedat quad filter set, and a CoolSnapHQ High Speed Monochrome CCD camera (Photometrics) run by Metamorph software." to "Images were acquired on a Nikon TE-2000 microscope using a 1.45 NA 100x objective, Sedat quad filter set, PIFOC Z-axis focus drive (Physik Instruments, Cranfield, UK), and a CoolSnapHQ High Speed Monochrome CCD camera (Photometrics) run by Metamorph software. This provided a pixel size of 0.0645 µm2. For general widefield images shown in figures 1B, 3B, 6B, 7A and 7B, the image was taken from a focus point at the midplane of the nucleus as this generally excludes signal from any other staining above or below the nucleus and generally affords the widest view of the ER as well outside the nucleus. For Figure 3E images, image stacks (0.2 ÎĽm steps) were deconvolved using AutoquantX (Media Cybernetics, UK). Images in Figure 6A were taken using a Zeiss LSM510 Meta confocal microscope. Micrographs were saved from source programs as 12-bit.tif files and analyzed with Image Pro Plus software and/or prepared for figures using Photoshop CS6."

Reviewer 2 Report

In their manuscript, Saiz-Ros et al. identify the vesicle fusion protein (VFP) vesicle-associated membrane protein-associted protein B (VAPB) as a host protein required for the nuclear egress of herpes simplex virus type-1 (HSV-1) capsids. While the HSV-1 proteins pUL31 and pUL34 are critical mediators of the egress of capsids from the nucleoplasm into the cytoplasm through the nuclear envelope, several other viral proteins including pUS3, gB, and gH also contribute to this process. Here, the authors test their hypothesis that host vesicle fusion proteins present within the nuclear envelope facilitate HSV-1 capsid primary envelopment at the inner nuclear membrane and/or de-envelopment fusion with the outer nuclear membrane. To begin to do so, Saiz-Ros et al. first employed a proteomics-based approach to search for candidate host proteins that might facilitate the fusion of primary enveloped virions within the perinuclear space of the nuclear envelope with the outer nuclear membrane. Since inner nuclear membrane proteins that become part of the primary envelope and outer nuclear membrane proteins that promote de-envelopment fusion are likely to increase in abundance in the endoplasmic reticulum of infected cells, the authors searched for host proteins that could be found in the nuclear envelope of uninfected cells, which were increased in the endoplasmic reticulum of infected cells relative to uninfected cells. Using this approach, they identified a subset of host vesicle fusion proteins as possible de-envelopment fusion facilitators, with VAPB being the most enriched of these proteins. Their investigation into the potential role of VAPB during HSV-1 replication revealed that VAPB-depletion significantly reduced viral titers, increased the accumulation of nuclear capsids and primary enveloped virions as well as decreased the number of cytoplasmic capsids. VAPB was also demonstrated to associate with primary enveloped virions and with capsids within the nucleoplasm, suggesting that VAPB might directly participate in HSV-1 nuclear egress as part of a larger complex together with viral proteins. Overall, the data and their interpretation presented within this well-written manuscript are compelling and deserving of publication as an article in Cells. However, there are several minor issues that need to be resolved before this manuscript can be accepted for publication. These issues are presented below:

1)    It would be very helpful if the authors were to include information about the protein domain organization of and known functions performed by VAPB in the Results section.

2)    Given that VAPB has a C-terminal transmembrane domain, it is unclear to me why it would be able to associate with nucleocapsids. How do the authors envision that this is occurring?   

3)    In the Fluorescence Microscope section of the Materials and Methods, there is no information provided regarding the fluorescent dyes that are conjugated to the secondary antibodies used for immunofluorescence staining and microscopy. Please include this information. 

4)    In the Fluorescence Microscope section of the Materials and Methods, the dilutions of the primary antibodies used for Western blotting or immunofluorescence staining are not provided, whereas the dilution of DAPI used is. Please provide this information. 

5)    Regarding the NE:ER quantification method presented in the Fluorescence Microscope section of the Materials and Methods, it is unclear what the authors mean by the statement “taking 5 pixels in either direction to define NE fluorescence intensity”. They should provide the size of a pixel given their imaging set-up (i.e. camera, objective, and any additional magnification lens if applicable).

6)    It would be helpful if the authors were to provide the positions of molecular weight standards on all of the Western blots presented in the manuscript. 

7)    The Western blots presented in Fig. 1A for pUS3 and gC are overly contrasted. They should adjust the contrast of these blots, and all of the other blots presented within the manuscript, so that some their background can be observed.

8)    The authors need to better define “dNSAF” and how it is used to determine the abundance ratio of proteins in mass spectrometry experiments. This would help the reader to understand how to interpret the plots presented in Figs. 2A-B as well as why the authors chose not to examine the role of VAMP7 during HSV-1 nuclear egress.

9)    Would it be useful to include a plot of the dNSAF HSV-1 NEs/mock NEs?

10)The images of cells stained for VAPB presented in Fig. 3B look quite different from those presented in Fig. 3E. Were these images processed differently from each other in some way?

11)The authors need to better describe how the data presented in Fig. 3D were generated. In the figure legend, they mention that “mean fluorescence intensities from whole NE and ER in sections were quantified”, yet this does not clearly explain what was done.

12)The figure legend for Fig. 4 needs to clearly state that the viral titer experiments presented in panels C and D were performed at 16 hpi.

13)It is unclear why viral titers were measured via plaque assays performed in U2OS cells when the rest of the experiments presented in this manuscript were performed in HeLa cells. Can the authors provide clarification here?

14)Regarding Fig. 4E, are there any statistics that were performed to determine whether or not the viral titers measured at the different time points were different between siControl and siVAPB?

15)Regarding Fig. 5, the authors should provide some arrows to draw their reader’s attention to exactly what they want them to be looking at in these EM images. Moreover, is there a reason why Fig. 6 is not a part of Fig. 5?

16)The authors need to define what siNT is.

17)At which hpi were the images presented in Fig. 7A taken? Based on my reading of the text, I believe that they were taken at 16 hpi. If this were true, it is very strange to me that there is so little VP5 staining present within the nucleoplasm of the controls given the images presented in Fig. 1B, where there is a considerable amount of VP26-RFP-labeled capsids within the nucleoplasm.

18)It would be helpful if the authors could include representative images of the data presented within the plots provided in Figs. 7D-E.

19)Is there a difference between the two sets of images taken at 8 hpi in siNT- or siVAPB-treated cells presented in Fig. 8A or the two sets of images taken siNT- or siVAPB-treated cells presented in Fig. 8B?

20)What are the “PEPs” referenced in Fig. 9? Can the authors define this term, which I believe might be primary enveloped particles?

21)The authors mention within the Discussion that VAPB’s “role in calcium homeostasis could lead to indirect effects on the fusion step”. Could they provide some more explanation for how calcium homeostasis might influence primary enveloped virion de-envelopment fusion with the outer nuclear membrane?

Author Response

Responses are in blue underneath each reviewer comment

Comments and Suggestions for Authors

In their manuscript, Saiz-Ros et al. identify the vesicle fusion protein (VFP) vesicle-associated membrane protein-associted protein B (VAPB) as a host protein required for the nuclear egress of herpes simplex virus type-1 (HSV-1) capsids. While the HSV-1 proteins pUL31 and pUL34 are critical mediators of the egress of capsids from the nucleoplasm into the cytoplasm through the nuclear envelope, several other viral proteins including pUS3, gB, and gH also contribute to this process. Here, the authors test their hypothesis that host vesicle fusion proteins present within the nuclear envelope facilitate HSV-1 capsid primary envelopment at the inner nuclear membrane and/or de-envelopment fusion with the outer nuclear membrane. To begin to do so, Saiz-Ros et al. first employed a proteomics-based approach to search for candidate host proteins that might facilitate the fusion of primary enveloped virions within the perinuclear space of the nuclear envelope with the outer nuclear membrane. Since inner nuclear membrane proteins that become part of the primary envelope and outer nuclear membrane proteins that promote de-envelopment fusion are likely to increase in abundance in the endoplasmic reticulum of infected cells, the authors searched for host proteins that could be found in the nuclear envelope of uninfected cells, which were increased in the endoplasmic reticulum of infected cells relative to uninfected cells. Using this approach, they identified a subset of host vesicle fusion proteins as possible de-envelopment fusion facilitators, with VAPB being the most enriched of these proteins. Their investigation into the potential role of VAPB during HSV-1 replication revealed that VAPB-depletion significantly reduced viral titers, increased the accumulation of nuclear capsids and primary enveloped virions as well as decreased the number of cytoplasmic capsids. VAPB was also demonstrated to associate with primary enveloped virions and with capsids within the nucleoplasm, suggesting that VAPB might directly participate in HSV-1 nuclear egress as part of a larger complex together with viral proteins. Overall, the data and their interpretation presented within this well-written manuscript are compelling and deserving of publication as an article in Cells.

We thank the reviewer for their positive and supportive comments.

However, there are several minor issues that need to be resolved before this manuscript can be accepted for publication. These issues are presented below:

1)    It would be very helpful if the authors were to include information about the protein domain organization of and known functions performed by VAPB in the Results section.

We have added to Figure 2 a schematic of VAPB domain structure and described its structure/ function in the text by adding the following sentences to the beginning of section 3.3: "VAPB is a type II integral membrane protein previously characterized in the ER. Together with family member VAPA, it is thought to function with cytoplasmic vesicle transport proteins and cytoskeletal elements to maintain membrane structure and facilitate lipid transport, membrane trafficking and membrane fusion (45). VAPB is a C-terminally anchored protein with its primary mass facing the cytoplasm in the ER (nucleoplasm for the inner nuclear membrane population) (46). The N-terminal region has a major sperm protein (MSP) homology domain that interacts with FFAT motif (two phenylalanines in an acidic track) proteins (47) followed by a coiled-coil domain before the transmembrane segment (Fig. 2F). A mutation (P56S) has been reported in the VAPB MSP domain causing an autosomal dominant form of amyotrophic lateral sclerosis (ALS8) that results in VAPB aggregation and neurotoxicity (48)."

2)    Given that VAPB has a C-terminal transmembrane domain, it is unclear to me why it would be able to associate with nucleocapsids. How do the authors envision that this is occurring?  

The membrane topology of VAPB is actually consistent with its interacting with nucleocapsids as, being a C-terminally anchored type II transmembrane protein, the majority of the molecule will face the cytoplasm when in the ER and the nucleoplasm when in the inner nuclear membrane.  Thus, it will be fully accessible to bind any virus capsid or tegument proteins of the nucleocapsid with which it is capable of interacting.  However, the topology of insertion means that in theory the primary role for VAPB would be in envelopment as opposed to de-envelopment/fusion with the ONM because the VAPB on the ONM would be mostly facing the cytoplasm and without facing the lumen it would be unlikely to interact in any meaningful way to initiate fusion of the primary virion envelope with the ONM.  Nonetheless, it would be possible for VAPB in the ONM to interact with proteins on the inside of the primary envelope once initial fusion was initiated to stabilize the egress complex or that it would be part of a greater complex where in response to complex components facing the lumen that interact with the primary envelope VAPB would interact with other components in the ER/ONM.  There is one other possibility, which we do not deem likely, in that VAPB could alter its topology during infection.  The observation of gold particles conjugated to antibodies that recognize the N-terminal part of the protein in the nucleoplasm, cytoplasm and nuclear envelope lumen could be consistent with this possibility; however, we performed protease sensitivity experiments to test VAPB topology and did not see evidence of it assuming two different topologies.  As there are still other possible explanations for the results of these assays we decided it was best to not include the data until we can also use other approaches to unequivocally demonstrate VAPB topology in both mock-infected and HSV-1 infected cells.

            As far as the discussion goes, we had initially written "VAPB was observed to associate with nucleoplasmic virions in a membranous vesicle, suggesting a role in primary envelopment; however, the immunogold electron microscopy data indicates that VAPB localizes both to the ONM and the INM both before and during HSV-1 infection, making it present to potentially contribute to both steps. As the principal functional mass of VAPB should be facing the nucleoplasm in the INM and the cytoplasm in the ONM, a role in primary envelopment is easy to visualize, especially if VAPB can bind any capsid or tegument proteins. The lack of clarity in the composition of primary enveloped particles allows for a very wide range of potential partners, so this may be a significant undertaking; however, it will be important to test in the future. A role in de-envelopment seems less likely with ONM VAPB mass facing the cytoplasm as it would need to face the lumen in order to interact with luminal primary enveloped particles. However, once the initial fusion has begun, VAPB either in the primary envelope or in the ONM could be positioned to contribute to stabilize and complete the process." We have now modified this text to perhaps be more clear as "Immunogold EM labelling revealed VAPB in association with nucleoplasmic/ luminal virions in a membranous vesicle and, together with the nucleoplasmic accumulation of virus particles, this suggests a role in primary envelopment. As VAPB is in both the INM and ONM according to the immunogold EM data, it could potentially contribute to both primary envelopment and de-envelopment; however, the topology of VAPB is such that its principal functional mass should be facing the nucleoplasm in the INM and the cytoplasm in the ONM. Thus, it is perfectly poised to potentially interact with virus nucleocapsids during primary envelopment, but it should not be positioned to interact with the primary enveloped particles in the nuclear envelope lumen in order to initiate fusion with the ONM for de-envelopment unless, as has never been tested, it assumes a different topology during HSV-1 infection. Nonetheless, once the initial fusion has begun, VAPB either in the primary envelope or in the ONM could potentially be positioned to contribute to stabilizing and/or completing the process of nuclear egress."

3)    In the Fluorescence Microscope section of the Materials and Methods, there is no information provided regarding the fluorescent dyes that are conjugated to the secondary antibodies used for immunofluorescence staining and microscopy. Please include this information. 

We have changed this from "Following 1 h incubation with secondary antibodies (Jackson Laboratories) and 4’,6-diamidino-2-phenylindole (DAPI; 1:2000) in blocking buffer and washing coverslips were mounted with Vectashield." to "Following 1 h incubation with Alexa-fluor secondary antibodies (Jackson Laboratories; 1:1000) and 4’,6-diamidino-2-phenylindole (DAPI; 1:2000) in blocking buffer and washing coverslips were mounted with Vectashield. All secondary antibodies were donkey minimal cross-reactive conjugates to Alexa-fluor dyes matched to the Sedat quad filters used (488, 568, and 647)  to prevent signal bleedthrough and negative controls were generally done to further confirm lack of cross-reactivity."

4)    In the Fluorescence Microscope section of the Materials and Methods, the dilutions of the primary antibodies used for Western blotting or immunofluorescence staining are not provided, whereas the dilution of DAPI used is. Please provide this information. 

We have now included this information, changing the text to read: "Primary antibodies were applied for 1 h at RT: VAPB (66191-1, mouse monoclonal, Proteintech; 1:200), US3 (rabbit polyclonal; 1:500) and pUL34 (rabbit polyclonal; 1:500)  — both rabbit antibodies were kindly provided by Thomas Mettenleiter."

5)    Regarding the NE:ER quantification method presented in the Fluorescence Microscope section of the Materials and Methods, it is unclear what the authors mean by the statement “taking 5 pixels in either direction to define NE fluorescence intensity”. They should provide the size of a pixel given their imaging set-up (i.e. camera, objective, and any additional magnification lens if applicable).

Re the pixel size, we have now added this (0.0645 µm2) to the camera setup description in the Methods section that immediately precedes the statement in question here. As far as the approach goes, the principle is that the outermost edge of the DNA will be probably 10-50 nm from the inner nuclear membrane due to the lamin polymer and many chromatin-binding nuclear membrane proteins. There is an additional 50 nm spacing between the inner and outer nuclear membranes. Between that the DAPI signal might be further in due to the greater intensity with the predominant DNA mass and that the ER would only start at least 50 nm further out we have set 5 pixels that comes to roughly 300 nm to ensure that nuclear envelope signal does not contaminate measurements of ER signal. It is possible that a small percentage of the most proximal ER cisternae could contaminate the nuclear envelope signal, but this will be only a small percentage of the total ER signal and considering the intensity of the nuclear envelope signal and that the concentrated fluorescence intensity will extend for several pixels due to the resolution limit of widefield microscopy being between 100-200 nm it seemed the best compromise.  To better clarify this we have added the sentence: "Given the 100-200 nm limit of resolution for standard (non-super resolution) fluorescence microscopy and the architecture of the NE, this 300 nm distance from the edge of the DNA signal should capture the full NE signal."

6)    It would be helpful if the authors were to provide the positions of molecular weight standards on all of the Western blots presented in the manuscript. 

We provided the positions of molecular weight standards on all gels showing a range of protein molecular weights or the whole gel (Fig. 1C, 3A) and have now also shown the actual bands of the standards themselves for Fig. 1C.  We did not provide them in the cases where we show just the short stretch of the gel where a particular protein was highlighted by Western because in each case this is a unique band of the expected molecular weight.  We have considered this particular request to add the positions of the markers for these short stretches and think that this would actually be confusing to most readers as in some cases there would be no molecular weight standards in the selected region of the gel and in others there might be two. This would also be confusing from the standpoint that when molecular weights are given for selected regions they typically just give the molecular weight of the protein of interest and not of the standards surrounding it. However, we have instead added the estimated molecular weights of the bands stained by the antibodies from the gels.

7)    The Western blots presented in Fig. 1A for pUS3 and gC are overly contrasted. They should adjust the contrast of these blots, and all of the other blots presented within the manuscript, so that some their background can be observed.

We have gone back to the original files (Li-COR for the pUS3 and an ECL film for the gC) and improved this as best as we could without modifying grey levels.  Unfortunately, one of the blots was really incredibly clean.

8)    The authors need to better define “dNSAF” and how it is used to determine the abundance ratio of proteins in mass spectrometry experiments. This would help the reader to understand how to interpret the plots presented in Figs. 2A-B as well as why the authors chose not to examine the role of VAMP7 during HSV-1 nuclear egress.

In the original Methods section we had written: "To estimate relative protein levels, distributed normalized spectral abundance factors (dNSAFs) were calculated for each non-redundant protein (Table 1 and Table S1), as described in (33)". This has now been expanded to: "Spectral-count based label free quantitation was used to estimate the relative levels of the proteins detected by mass spectrometry in each sample.  In shotgun proteomics, the frequency of peptides being fragmented by the mass spectrometer (spectral counts) correlates with the abundance of the proteins these peptides derive from.  Because longer proteins tend to generate more tryptic peptides, spectral counts are normalized by the protein molecular weight or length, defining a “spectral abundance factor” (SAF), which is further normalized against the sum of SAFs calculated for each protein/protein group detected in a sample.  To deal with peptides shared between multiple proteins, distributed Normalized Abundance Factors (dNSAFs) are calculated for each non-redundant protein/protein group (Table 1, 2 and Table S1), as described in (33):

 with ,

in which shared spectral counts (sSpC) are distributed based on spectral counts unique to each protein i (uSpC) divided by the sum of all unique spectral counts for the M protein isoforms that shared peptide j with protein i."

9)    Would it be useful to include a plot of the dNSAF HSV-1 NEs/mock NEs?

That would actually be impossible because we did not do HSV-1 infected NEs due to the fact that the lamin A phosphorylation by the virus makes the NEs very weak so that they cannot be cleanly isolated.  This is the reason we focused on isolating MMs from the infected cells and then testing host proteins that were in the NE in mock-infected cells so that we knew they were present before viral infection and then tested how their abundance changed in the MMs upon infection.

10)The images of cells stained for VAPB presented in Fig. 3B look quite different from those presented in Fig. 3E. Were these images processed differently from each other in some way?

The images in Fig. 3B are widefield while those in Fig. 3E are deconvolved from Z-stacks.  We can see now that we had not phrased this clearly before and have now modified the figure legend to make it clear that only the images in 3E and not those in 3B were deconvolved.

11)The authors need to better describe how the data presented in Fig. 3D were generated. In the figure legend, they mention that “mean fluorescence intensities from whole NE and ER in sections were quantified”, yet this does not clearly explain what was done.

We further described how the NE and ER were determined in the Methods as noted in response to point 5 above. We have additionally modified the text in the figure legend to state: "...after defining the NE in relation to the DAPI signal the total NE fluorescence and all fluorescence signal outside the nucleus was quantified. From this data mean fluorescence intensities from the whole NE and ER in sections were quantified, the ratios of NE:ER signal were determined, and their distribution was plotted using a log scale."

12)The figure legend for Fig. 4 needs to clearly state that the viral titer experiments presented in panels C and D were performed at 16 hpi.

We have now added "and supernatant virions also collected at 16 hpi" to the 4C legend. For 4D the text already stated "and cell pellets collected at 16 hpi."

13)It is unclear why viral titers were measured via plaque assays performed in U2OS cells when the rest of the experiments presented in this manuscript were performed in HeLa cells. Can the authors provide clarification here?

This is standard for virus studies: titers are measured in a cell type most optimized for virus entry so as to better gauge the actual number of infectious particles. U2OS cells are used for HSV-1 because they are great for viral entry and replication. We have now added a sentence to the Methods at the end of 2.6 to explain this for non-virologist readers.

14)Regarding Fig. 4E, are there any statistics that were performed to determine whether or not the viral titers measured at the different time points were different between siControl and siVAPB?

Unfortunately we have only been able to carry out this experiment once because the VAPB siRNA appears to have gone bad and new material was delayed due to the holidays. Thus, as we did not have biological replicates we did not generate error bars.

15)Regarding Fig. 5, the authors should provide some arrows to draw their reader’s attention to exactly what they want them to be looking at in these EM images. Moreover, is there a reason why Fig. 6 is not a part of Fig. 5?

We have now added arrows to the figure as requested to highlight the different structures noted in the text and legend.

Re the second point, we had initially separated these for the aesthetic reason that the quantification took a much much smaller space than the images and it made for wasted white space in the figure. However, the reviewer is right that this may be confusing as the data belong together and so we have stretched out the table to better fill the white space and fused the two figures.

16)The authors need to define what siNT is.

We actually had defined this in the legend as "non-target"; however, we can see now that this is confusing because we had used "control" for the same thing in the other figures.  We have thus changed it now to "control".

17)At which hpi were the images presented in Fig. 7A taken? Based on my reading of the text, I believe that they were taken at 16 hpi. If this were true, it is very strange to me that there is so little VP5 staining present within the nucleoplasm of the controls given the images presented in Fig. 1B, where there is a considerable amount of VP26-RFP-labeled capsids within the nucleoplasm.

To remove any confusion we have now stated in the figure legend that they were taken at 16 hpi.

Regarding the issue of the VP5 staining in the nucleoplasm, the staining is actually there but was masked by the strong DAPI signal from the DNA staining in the merged images. Since we had taken all images using the same settings this was weaker compared to the signal from the VAPB siRNA due to the accumulation of virus particles in the nucleoplasm. We had orignally tried to keep the intensity from appearing saturated in the VAPB siRNA images, which led to the masking of the signal by the DAPI. We have both increased the intensity a bit and now show separately beneath the merged image one with just the VP5 staining where the nuclear limit defined by the DAPI staining in the other channel is demarcated by a red line. This now clearly shows that there is VP5 signal also in the nucleoplasm in the control cells.

18)It would be helpful if the authors could include representative images of the data presented within the plots provided in Figs. 7D-E.

The images in panel B are exactly the images that were used for the quantifications in both panels C and D. As indicated in panel D roughly 50 different cell images for each condition were analyzed for the quantification. We have added to the figure legend to better clarify both how the analysis was done and make clear the links between experiments.

Panel E is from a different experiment and when we are processing cells for qPCR we generally do not also image the experiment. Thus, there are no images associated with this experiment. However, all these experiments were performed in the same manner with the same cells, HSV-1 wild-type strain, MOI, and hpi.

19)Is there a difference between the two sets of images taken at 8 hpi in siNT- or siVAPB-treated cells presented in Fig. 8A or the two sets of images taken siNT- or siVAPB-treated cells presented in Fig. 8?

In a way it is kind of sad that when one shows multiple images in a manuscript it generates the assumption that one set must have been mislabeled. It would be better if as a practice scientists started showing multiple images, particularly as the community is beginning to realize that population variation is relevant. The images are as labeled both reflecting cells at 8 hpi. The reason two cells are shown is so that the reader can observe both more rounded cells with altered nuclei and more flattened cells without major nuclear disruption within the popultion. Since we expect this might be a common misperception on the part of readers, we have specifically stated this now in the figure legend, adding: "Two separate images are shown to highlight that an increased concentration of staining at the NE (defined as the edge of the DNA staining) is observed in both flatter cells with more normal looking nuclei and more rounded cells with distorted nuclei within the same population."

20)What are the “PEPs” referenced in Fig. 9? Can the authors define this term, which I believe might be primary enveloped particles?

The reviewer is correct and we have now defined this in the figure legend.

21)The authors mention within the Discussion that VAPB’s “role in calcium homeostasis could lead to indirect effects on the fusion step”. Could they provide some more explanation for how calcium homeostasis might influence primary enveloped virion de-envelopment fusion with the outer nuclear membrane?

Reviewer 3 Report

Saiz-Ros N et al. presented two interesting finding;(1) Silencing of VAPB, a known vesicle-associated membrane protein B significantly reduced cell-associated and cell-released virus titer , and  (2) VAPB is an important player in exit of primary enveloped HSV-1 virions from the nucleus. These findings are confirmed using siRNA's silencing, presence of VAPB  in NEs and MM's was identified using mass-spectrometry ,immunogold labeling  electron  microscopy, immunofluorescence studies,  FISH  and PCR. The work appears to be thorough and paper reads well . However, there are some major  and minor points that should be clarified.

 Major concerns:

1. One of the most important finding is the accumulation of viral particles in nuclear and peri-nuclear  membranes  in VAPB knockdown cells (Figure 7). Result of   immunofluorecent studies using Abs to major capsid protein Vp5 presented in Figure 7A looks convincing  but cytoplasmic staining will be helpful to confirm the localization of viral particles. Results of FISH in Figure 7B using ICP27 viral gene as a probe for FISH need to be confirmed with more images of nuclei taken at the different slices of cells. Nucleus in siVAPB silenced cells looks   smaller in size than the same nucleus in siRAB24 or siRAb knockdown cells. Quantifying this data of multiple images of the same nucleus would increase confidence in the conclusion.

 2 .Experiment using immunogold labeling technique is informative but requires some biochemical conformation such as a co-immunoprecipitation of VAPB with viral proteins.

 Minor concerns:

1.     Figure 3E: it was not clear what is a difference between three panels. There are no labels on left side of images. Figure 3E also requires some negative control such as mock/infected cells and staining with antibodies to viral proteins that do not involve into nuclear envelopment.  

2.     Figure 4A: silencing by siRab24 accessed by western blot is not convincing, requires a quantification. Please, indicate the molecular weights of proteins.

3.     Figure 8B: no labels on the left side of panels.  

Author Response

Responses are in blue below each reviewer comment

Comments and Suggestions for Authors

Saiz-Ros N et al. presented two interesting finding;(1) Silencing of VAPB, a known vesicle-associated membrane protein B significantly reduced cell-associated and cell-released virus titer , and  (2) VAPB is an important player in exit of primary enveloped HSV-1 virions from the nucleus. These findings are confirmed using siRNA's silencing, presence of VAPB  in NEs and MM's was identified using mass-spectrometry ,immunogold labeling  electron  microscopy, immunofluorescence studies,  FISH  and PCR. The work appears to be thorough and paper reads well . However, there are some major  and minor points that should be clarified.

 Major concerns:

1. One of the most important finding is the accumulation of viral particles in nuclear and peri-nuclear  membranes  in VAPB knockdown cells (Figure 7). Result of   immunofluorecent studies using Abs to major capsid protein Vp5 presented in Figure 7A looks convincing but cytoplasmic staining will be helpful to confirm the localization of viral particles.

We have done other cytoplasmic co-stainings with both ER markers (calnexin in Fig. 3) and the Golgi (GM-130 in Fig. 7, prev 8); however, we did not co-stain with a marker for cytoplasmic staining in the VP5 experiment. Nonetheless, in this experiment the cells were washed prior to fixation and therefore most virus particles that had not yet infected cells would likely have been washed away along with any released fresh progeny virus, though the 16 hpi timepoint would be early to have much released progeny virus. Therefore, the VP5 signal outside the nuclear limits as defined by the DAPI staining of DNA is likely all contained within the cytoplasm. At the same time, even if the signal outside the nucleus included lots of progeny virus released, this would only further emphasize the observed accumulation of virus particles in the nucleoplasm in the VAPB knockdown cells, especially as in experiments in Fig. 4B and C we showed the significant reduction in virus titers using collected supernatant virions i.e. the virus is not being released in the knockdown.

To allow the reviewer and readers to better distinguish nuclear from cytoplasmic VP5 staining we have added an additional panel showing just the VP5 staining beneath the merged images in which the nuclear envelope is defined by a line marking the edge of the DAPI signal in the other channel.

Results of FISH in Figure 7B using ICP27 viral gene as a probe for FISH need to be confirmed with more images of nuclei taken at the different slices of cells. Nucleus in siVAPB silenced cells looks   smaller in size than the same nucleus in siRAB24 or siRAb knockdown cells. Quantifying this data of multiple images of the same nucleus would increase confidence in the conclusion.

Our lab as well as that of Wendy Bickmore and many others have published many papers on genome organization where we show that there is no change in the directionality of results and typically only minor changes in intensity between analyzing cells by FISH using a mid-section through the nucleus and performing 3D analysis when ~50 cells are analyzed (e.g. see Zuleger et al., 2013 Genome Biol). As the processing time for the 3D analysis is considerably greater than for the 2D mid-section analysis we have elected to use the 2D analysis here.

Although when we have co-stained with ER markers, we typically observe the preponderance of ER staining also in the midplane sections, we nonetheless acknowledge that disruption to the ER and Golgi during herpesvirus infection as well as rounding up of cells at later stages of infection could in theory skew these results. Accordingly, we have both emphasized now in the figure legend that the cells were analyzed by a 2D rather than a 3D approach and we have added a caveat to the results: "Images for this analysis were taken from the nuclear midplane where nuclear area is greatest. While generally the ER sectional area is also greater at the nuclear midplane, it is possible that changes induced by the HSV-1 infection that alter both ER and Golgi membrane distribution might yield a skewed distribution for the positioning of virus particles, Nonetheless, all samples were treated equally and so the differences between the VAPB knockdown and control samples should still be relevant."

 2 .Experiment using immunogold labeling technique is informative but requires some biochemical conformation such as a co-immunoprecipitation of VAPB with viral proteins. 

A new student has spent much of the past month trying to co-immunoprecipitate partners of VAPB in response to this comment. He can pull some VAPB down, but does not see a clear interaction with pUL34. Unfortunately the secondary antibody recognizes a band of the same molecular weight that could mask an interaction so that we cannot definitively say that no interaction occurs, though my opinion is that the lack of a difference between VAPB and control antibodies in staining at the molecular weight for pUL34 in the coIP lane that was cut down the middle in the gel shown argues against an interaction between VAPB and pUL34. Nonetheless, the problems other labs have had in adding tags to pUL34 and in coIP'ing partners with pUL34 suggest that occupancy of binding sites on the protein that block antibody and partner protein binding at the same time could mask relevant interactions. All in all, to identify partner proteins for VAPB is a major undertaking and goes well beyond the remit of this first report demonstrating that knockdown of VAPB yields an effect on nuclear egress.

 Minor concerns:

1.     Figure 3E: it was not clear what is a difference between three panels. There are no labels on left side of images. Figure 3E also requires some negative control such as mock/infected cells and staining with antibodies to viral proteins that do not involve into nuclear envelopment.

There are no labels on the side for this figure because it is showing three different fields from the same condition and experiment. 

2.     Figure 4A: silencing by siRab24 accessed by western blot is not convincing, requires a quantification. Please, indicate the molecular weights of proteins. 

We have added the quantification to the right side in panel A of the figure. 

Molecular weights for markers were shown for all full gels previously. We had not included the molecular weights for individual bands shown separately from the rest of the gel after Western blotting. We have now added the molecular weights of the bands next to these.

3.     Figure 8B: no labels on the left side of panels.  

There are no labels on the side because all the information is given across the top of the panels. While we could use the space on the empty side panels to put the siCntl and the siVAPB labels instead of at the edge on top, we think it is more clear to use the same structure as in panel A to demarcate the treatment. The B top and bottom panels are just two different images from the same condition. To make this clear we have added to the figure legend: "Top and bottom panels are different images from the same condition."

Reviewer 4 Report

Review

Cells Manuscript ID: cells-384762

Host Vesicle Fusion Protein VAPB Is Required for the Nuclear Egress Stage of Herpes Simplex Virus Type-1 (HSV-1) Replication submitted by Natalia Saiz-Rosa et al.

Herpesviruses are unique in that during morphogenesis, nucleocapsids transit membranes twice, i) during primary envelopment to exit the nucleus, and ii) during secondary envelopment to form mature infectious particles released  to the extracellular milieu. The authors of this manuscript postulated that vesicle fusion proteins present in the nuclear envelope might facilitate primary envelopment of nucleocapsids and/ or their de-envelopment with the outer nuclear membrane.

To identify novel host factors involved in nuclear egress, the authors prepared nuclear envelopes (NE) and microsomal membranes (MM) in presence and absence of Herpes simplex virus type 1 (HSV-1) infection and determined their protein composition using mass spectrometry. Several vesicle fusion proteins (VFPs) were identified among which the vesicle-associated membrane protein associated protein B (VAPB) was most enriched. To validate this finding, the authors followed the subcellular localization of VAPB in the course of infection in comparison with the nuclear egress protein pUL34, the viral kinase pUS3 as well as cellular markers for ER and Golgi. RNAi of VAPB reduced both the cell-associated as well as the extracellular virus titer. Furthermore, depletion of VAPB increased the levels of nucleocapsids in the nucleus and at the same time reduced the number of cytoplasmic capsids. Immunogold-labeling electron microscopy detected VABP at nuclear membranes and in association with primary enveloped HSV-1 particles.

While the authors provide a large body of data, overall, they do not support the major conclusion, that VAPB is required for a the nuclear egress stage of HSV-1.

Major concerns:

-          RNAi and the virological data (Fig. 4) seem to support a specific role of VAPB during HSV-1 infection. However, direct evidence for a role of VAPB in nuclear egress is lacking:

o   VAPB is a protein found in various subcellular compartments including the ER and the Golgi compartment. In absence of infection, VABP shows a rather diffuse ER-like distribution that changed in the course of infection to a perinuclear localization resembling the Golgi compartment (Figure 3 B). A small, but statistically distinct increase of VAPB in the NE pool was described, however unlike stated, a distinct nuclear rim staining is not visible. Furthermore, the authors describe a partial recruitment of VAPB to the NE during infection, but point out the wide distribution of the NE:ER ratio between the mock and the 16 hpi condition. Altogether, no clear colocalization of pUL34 and VAPB or a recruitment of VAPB to the NE was found.

o   A protein involved in nuclear egress is expected to get in close contact with the nuclear egress complex (NEC). In order to support the conclusion that VAPB plays a role during nuclear egress, the authors should provide biochemical evidence for interaction between pUL34/pUL31 and VAPB, eg by co-purification of VAPB with the NEC.

-          Based on initial infection studies the authors chose to prepare MMs between 8 and 9 hpi at an MOI of 10. An MOI of 10 is rather high and likely accelerates the release of virions. IN accordance, the nuclear interior is almost empty of nucleocapsids and the MMs contain tegument proteins and viral glycoproteins typically expressed at a late step of infection. In addition, the pUL34 localization presented in Fig. 3 E as well as Fig. 8 A and B is not as reported during nuclear egress. At this stage of infection, pUL34 is primarily found at the NE and only later may diffuse to the ER. Altogether, the conditions chosen for infection are not suitable to report on nuclear egress / primary infection but rather on a late stage eg secondary envelopment.

-          Intense accumulations of nucleocapsids at the nuclear rim in general do not occur during wildtype virus infections. However, similar “assemblies” are frequently observed upon infection with recombinant viruses that express fluorescently tagged viral capsid proteins such as VP26-RFP. These kind of capsid proteins seem to aggregate in the nucleus forming intense spots of VP26-RFP that may be mistaken as accumulated nucleocapsids. Furthermore, improper tagging of VP26 was reported to impair nuclear capsid egress (Nagel et al., PLoS One 2012;7(8):e44177) contributing to the reduction in viral titers. Taken together, the viral infection system applied has strong limitations and is prone to artefacts.

-          As pointed out by the authors, the reduction in viral titers following RNAi could reflect a function of VAPB in nuclear egress or secondary envelopment. EM analysis showed that RNAi of VAPB resulted in more nuclear and less cytoplasmic virus particles. Experiments in Fig. 4 B (extracellular) and Fig. 4 D (cell-associated) suggest that in absence of VAPB, formation of intracellular / cytoplasmic virus is hampered (unfortunately, experiments are lacking that determine both cell-associated and extracellular virus in one experiment). Similar results may be expected for a role of VAPB in secondary envelopment. Therefore, the effects observed may potentially be indirect caused by congesting the trafficking paths of nucleocapsids from the nuclear interior to the cytoplasm.

Author Response

Responses are in blue beneath each reviewer comment

Comments and Suggestions for Authors

Review

Cells Manuscript ID: cells-384762

Host Vesicle Fusion Protein VAPB Is Required for the Nuclear Egress Stage of Herpes Simplex Virus Type-1 (HSV-1) Replication submitted by Natalia Saiz-Rosa et al.

Herpesviruses are unique in that during morphogenesis, nucleocapsids transit membranes twice, i) during primary envelopment to exit the nucleus, and ii) during secondary envelopment to form mature infectious particles released  to the extracellular milieu. The authors of this manuscript postulated that vesicle fusion proteins present in the nuclear envelope might facilitate primary envelopment of nucleocapsids and/ or their de-envelopment with the outer nuclear membrane.

To identify novel host factors involved in nuclear egress, the authors prepared nuclear envelopes (NE) and microsomal membranes (MM) in presence and absence of Herpes simplex virus type 1 (HSV-1) infection and determined their protein composition using mass spectrometry. Several vesicle fusion proteins (VFPs) were identified among which the vesicle-associated membrane protein associated protein B (VAPB) was most enriched. To validate this finding, the authors followed the subcellular localization of VAPB in the course of infection in comparison with the nuclear egress protein pUL34, the viral kinase pUS3 as well as cellular markers for ER and Golgi. RNAi of VAPB reduced both the cell-associated as well as the extracellular virus titer. Furthermore, depletion of VAPB increased the levels of nucleocapsids in the nucleus and at the same time reduced the number of cytoplasmic capsids. Immunogold-labeling electron microscopy detected VABP at nuclear membranes and in association with primary enveloped HSV-1 particles.

While the authors provide a large body of data, overall, they do not support the major conclusion, that VAPB is required for a the nuclear egress stage of HSV-1.

The reviewer is mis-stating our conclusions.  What we wrote in the abstract was "These data suggest that VAPB could be a cellular component of a complex that facilitates UL31/UL34/US3-mediated HSV-1 nuclear egress." The results end with the statement "suggests the possibility that VAPB participates in the process of primary envelopment." We repeatedly use words like "contributes" in the discussion and finish the discussion qualifying the role of VAPB to "involved". The only place where we used the term required was at the end of the introduction where we wrote "These data suggest that VAPB is required for the HSV-1 life cycle and may facilitate nuclear egress." This we have now toned down to "These data suggest that VAPB is an important contributor to the HSV-1 life cycle and may facilitate nuclear egress."

Major concerns:

-          RNAi and the virological data (Fig. 4) seem to support a specific role of VAPB during HSV-1 infection. However, direct evidence for a role of VAPB in nuclear egress is lacking:

·       VAPB is a protein found in various subcellular compartments including the ER and the Golgi compartment. In absence of infection, VABP shows a rather diffuse ER-like distribution that changed in the course of infection to a perinuclear localization resembling the Golgi compartment (Figure 3 B). A small, but statistically distinct increase of VAPB in the NE pool was described, however unlike stated, a distinct nuclear rim staining is not visible. Furthermore, the authors describe a partial recruitment of VAPB to the NE during infection, but point out the wide distribution of the NE:ER ratio between the mock and the 16 hpi condition. Altogether, no clear colocalization of pUL34 and VAPB or a recruitment of VAPB to the NE was found.

Again the reviewer is not accurately reflecting what is stated in the manuscript. We very clearly acknowledge that VAPB is largely in the ER throughout the manuscript and state for example that "In the mock-infected cells considerable ER distribution of VAPB was observed as well as a weak, but distinct rim around the nucleus and this staining was largely similar in infected cells (Figure 3B)." Regarding co-localization with pUL34 we stated "The majority of VAPB signal was in the ER while pUL34 concentrated in the nucleus, but both gave clear nuclear rim staining with this pool exhibiting partial overlap between the signals (Figure 3E). The data suggest a partial recruitment of VAPB to the NE during HSV-1 infection."

            Of note, since we began the process of trying to get this study published we have been contacted by two separate labs who have also found evidence for a separate pool of VAPB in the NE and the completely unbiased and independent IntAct database of protein interactions accessible on uniprot has reported an interaction between VAPB and the clearly inner nuclear membrane protein emerin. Thus the first criticism of VAPB at the NE is supported now not only by our first data, but by several separate labs and approaches.

            Regarding the issue of co-localization, we have now quantified the amount of overlapping signal and plotted the Pearson's Correlation Coefficient for both the NE signals and for the signals through the rest of the nucleus (now added to Figure 3E). This unbiased quantitative approach shows that there is a reasonable amount of overlap between the two signals and that there is more overlap for the pools of protein at the NE than in the rest of the cell.

·       A protein involved in nuclear egress is expected to get in close contact with the nuclear egress complex (NEC). In order to support the conclusion that VAPB plays a role during nuclear egress, the authors should provide biochemical evidence for interaction between pUL34/pUL31 and VAPB, eg by co-purification of VAPB with the NEC.

This comment seems to be based on an assumption that pUL31/34 are involved in everything that happens during nuclear egress and that they must interact with every other protein involved in any aspect of nuclear egress. Thus far pUL34 partners are quite limited, which, considering the complexity of vesicle trafficking in the cell and the propensity of herpesviruses to co-opt host pathways, is more likely to reflect that existing antibodies compete for binding sites with partners than that such partners do not exist. This is the more likely because of problems with pUL34 function when it is expressed as a fusion protein. Nonetheless, we did IP VAPB and test by Western for the possibility of pUL34 being a partner, as detailed above for reviewer 3. A blot is shown in the reviewer 3 responses. The result was negative, but also inconclusive for reasons also detailed in the response to reviewer 3.

-          Based on initial infection studies the authors chose to prepare MMs between 8 and 9 hpi at an MOI of 10. An MOI of 10 is rather high and likely accelerates the release of virions. IN accordance, the nuclear interior is almost empty of nucleocapsids and the MMs contain tegument proteins and viral glycoproteins typically expressed at a late step of infection. In addition, the pUL34 localization presented in Fig. 3 E as well as Fig. 8 A and B is not as reported during nuclear egress.

At this stage of infection, pUL34 is primarily found at the NE and only later may diffuse to the ER. Altogether, the conditions chosen for infection are not suitable to report on nuclear egress / primary infection but rather on a late stage eg secondary envelopment.

We are confused about what argument the reviewer is trying to make here.  The reviewer states that pUL34 is in the NE at the time we set to take the MMs ("At this stage of infection, pUL34 is primarily found at the NE ") and moreover we further confirm that this is the case in the 8 h images in Figure 7 (previously Figure 8). The timing chosen was precisely to capture cells in the early stages of nuclear egress when the virus is actively engaging with the NE.  Why then the reviewer subsequently states that these conditions are optimized to report "on a late stage eg secondary envelopment" is thus not clear. 

Regarding the localization of pUL34 at the NE and the 8 h timepoint, at the MOI=10 that we used for nearly all experiments in this study the other virus protein pUS3 that is known to contribute to nuclear egress had still not achieved full expression while gC1 was not expressed at measurable levels yet. Therefore, the 8 h timepoint shown does not reflect a late stage in infection. This argument might be able to be applied to some experiments at 16 hpi, but not the 8 hpi images shown in Figure 7A (previously 8A). If the staining is not as ideal as the reviewer is used to then it reflects some aspect of the antibodies provided by Thomas Mettenleiter, who is one of the leading researchers in this area. Moreover, the targeting to the NE is not the point of this figure, but rather the fact that the pUL34 staining pattern is similar between the control and VAPB knockdown conditions.

-          Intense accumulations of nucleocapsids at the nuclear rim in general do not occur during wildtype virus infections. However, similar “assemblies” are frequently observed upon infection with recombinant viruses that express fluorescently tagged viral capsid proteins such as VP26-RFP. These kind of capsid proteins seem to aggregate in the nucleus forming intense spots of VP26-RFP that may be mistaken as accumulated nucleocapsids. Furthermore, improper tagging of VP26 was reported to impair nuclear capsid egress (Nagel et al., PLoS One 2012;7(8):e44177) contributing to the reduction in viral titers. Taken together, the viral infection system applied has strong limitations and is prone to artefacts.

We did not use the VP26-RFP virus for MM preps nor any other experiment other than the experiment shown in Figure 1B where we used visual cues in infected cells as one of several criteria used to optimize the timing of when we would take the infected cells to prepare MMs.  All other experiments including other optimization experiments were performed with wild-type strain 17+.

In case this was not clear, we have now stated in Methods section 2.1 that this virus was only used for the experiment in Figure 1B.

-          As pointed out by the authors, the reduction in viral titers following RNAi could reflect a function of VAPB in nuclear egress or secondary envelopment. EM analysis showed that RNAi of VAPB resulted in more nuclear and less cytoplasmic virus particles. Experiments in Fig. 4 B (extracellular) and Fig. 4 D (cell-associated) suggest that in absence of VAPB, formation of intracellular / cytoplasmic virus is hampered (unfortunately, experiments are lacking that determine both cell-associated and extracellular virus in one experiment). Similar results may be expected for a role of VAPB in secondary envelopment. Therefore, the effects observed may potentially be indirect caused by congesting the trafficking paths of nucleocapsids from the nuclear interior to the cytoplasm.

We certainly agree that if there were just a demonstration of reduction in extracellular virus without a reduction in cell-associated genomes that this could be explained as readily by a defect in secondary envelopment as in nuclear egress. However, for the reviewer to make this statement it would be helpful if the reviewer would explain how an effect only in secondary envelopment for VAPB would result in virus particles accumulating in the nucleus and NE lumen and decreasing in the cytoplasm? If VAPB was only functioning in secondary envelopment the nucleocapsids should still be able to get through the nuclear envelope and would accumulate in the cytoplasm after de-envelopment. We can envision ways that a function of VAPB in the Golgi or ER could have an indirect effect on nuclear egress: 1) if VAPB were important for production of virus or host proteins needed either preparatory to or for nuclear egress; 2) if if VAPB were important for trafficking of virus or host proteins needed either preparatory to or for nuclear egress. In both cases this is separate from a VAPB role in secondary envelopment having an effect on nuclear egress. Regarding the first possible indirect mechanism we do not see a reduction in several virus proteins we tested (gC1, pUL34, ICP27, gD and VP26). The data for the first two has been added to Figure 4 in the new panel F. While this cannot be completely excluded until every virus and host cell protein involved in nuclear egress or capsid assembly has been tested, since we don't know every protein involved in these processes this is impossible. For the second possible indirect mechanism we do not see any difference in the distribution of pUL34 in Figure 7A (previous Figure 8A) between siControl and siVAPB conditions. The reviewer seems to have missed the point of this figure, which was not to show co-localization or only NE targeting of pUL34 but rather that the pUL34 distribution is not different between control and VAPB knockdown conditions. Nonetheless, to further qualify our results by elaborating this possibility, we have added to the discussion the clause: "however, it is also possible that HSV-1 disruption of ER and Golgi membranes unintentionally results in a redistribution of VAPB where it can more indireclty contribute to nuclear egress and it remains possible that the trafficking of some important player in nuclear egress besides pUL34 is not able to gain access to the NE in the VAPB knockdown cells."

Round 2

Reviewer 1 Report

Although many points remain to be clarified, this manuscript may lay the ground for further experimentation. However, even though the authors agreed to tone down their statement within the text (“This we have now toned down to "These data suggest that VAPB is an important contributor to the HSV-1 life cycle and may facilitate nuclear egress."”), the statement in the title still remains (VAPB is required for the nuclear egress of HSV-1). The author might think to tone down the title as well.

Author Response

Reviewer 1 Response: We think there may have been some confusion regarding which version of the text the reviewer downloaded as the version I just downloaded that was the submitted revision had the title as "Host vesicle fusion protein VAPB contributes to the nuclear egress stage of Herpes Simplex Virus Type-2 (HSV-1) replication".  As we could only change this in the revised text and did not have access to change this or to add the additional author on the submission site, it is possible that the reviewer is responding to the electronic title that is linked to the submission and which accordingly has not changed.  We have highlighted the change from "required" to "contributes to" in the revision 2 title in blue.

Reviewer 3 Report

The major criticism expressed by reviewer  was the lack of biochemical assays  to confirm the interaction of VAPB with viral proteins. Authors tried to perform co-immunoprecipitation  of VAPB with pUL34 but failed. Their failure is due to lack of specificity of used antibodies . That is a common problem of co-immunoprecipitation methods, but the biochemical approaches to show protein-protein interactions should be included in the future  studies. 

Author Response

Reviewer 3 Response: We thank the reviewer for acknowledging the difficulties of these experiments and noting that they are not needed for the current manuscript.  We can assure them that the new student we added to the paper is continuing trying to test for interactors with BioID, coIP, and 2-hybrid analyses as part of his project.

Reviewer 4 Report

A few changes are still necessary: The figures have been rearranged and extended, the sequence of numbering however has not been adjusted in the figure legends (Fig. 2-Fig. 6). This is also true for the manuscript, in particular the results section. The legend for Fig. 5 A (currently Fig. 4) should be adjusted: There is a new panel but left and right panels are not discriminated, description of the right panel is missing. I hope my comments support the final descision.

Author Response

Reviewer 4 Response: We think that much of the confusion here comes from that the copy editors in adding the figures to our text file physically placed Figure 6 between Figures 1 and 2; however, the figure legends themselves all matched the figures and the figures were referred to in the correct places within the text.  We have moved Figure 6 now to be placed between Figures 5 and 7 and have done a search for "fig" and carefully checked each reference throughout the text for accuracy and clarity.  Between this and the specific reviewer's comments, we did notice a few minor issues that we have changed in the revised text.

Figure 2: We think that the reviewer comment might have indicated that they thought it was wrong to refer to the additional panel D we added in the revision prior to panel C in the text.  Therefore we have revised the figure to switch the order of panels D and C so that they reflect the order they are mentioned in the text and accordingly fixed the figure references in the text.  We have highlighted the changes for this in blue in the attached text.

Figure 4: We think what the reviewer was referring to here is that in panel A we added the quantification to the side of the Western images and did not note this in the legend. We have now added to the legend: "Quantification from three experiments is given in the right panel: error bars are to the standard deviation and the p-values using a one-way ANOVA and multiple comparisons by Dunnett's test for comparing each condition with the control are shown (**p≤0.01; ****p≤0.0001)." This has also been highlighted in blue.

Figure 5: To better clarify which panels are being referred to we added "upper panels" to the first figure reference and after the second sentence referring to the same panel additionally added "(Figure 5A, lower panels)". Again the changes to the text are highlighted in blue.

Figure 8: We noticed that we had not updated the text when we added an extra panel D and changed the previous D to E.  This had been modified in the figure legend, but the text still just referred to panel D.  We have made minor changes to both the text and legend to better explain what is shown in each panel and marked the changes with blue highlighting.